# EXPLAINING THE OUT-OF-DISTRIBUTION DETECTION PARADOX THROUGH LIKELIHOOD PEAKS

## ABSTRACT

Likelihood-based deep generative models (DGMs) commonly exhibit a puzzling behaviour: when trained on a relatively complex dataset, they assign higher likelihood values to out-of-distribution (OOD) data from simpler sources. Adding to the mystery, OOD samples are never generated by these DGMs despite having high likelihoods. This two-pronged paradox has yet to be conclusively explained, making likelihood-based OOD detection unreliable. Our primary observation is that high-likelihood regions will not be generated if they contain minimal probability mass, which can occur if the density is sharply peaked. We demonstrate how this seeming contradiction of large densities yet low probability mass can occur around data confined to low dimensional manifolds. We also show that this scenario can be identified through local intrinsic dimension (LID) estimation, and propose a method for OOD detection which pairs the likelihoods and LID estimates obtained from a pre-trained DGM. Moreover, we provide an efficient method for estimating LID from a normalizing flow model, improving upon existing estimators, and enabling state-of-the-art OOD detection performance with respect to comparable flow-based benchmarks.

## 1 INTRODUCTION

Out-of-distribution (OOD) detection (Quiñonero-Candela et al., 2008; Rabanser et al., 2019; Ginsberg et al., 2022) is crucial for ensuring the safety and reliability of machine learning models, given their deep integration into real-world applications ranging from finance (Sirignano & Cont, 2019) to medical diagnostics (Esteva et al., 2017). In areas as critical as autonomous driving (Bojarski et al., 2016) and medical imaging (Litjens et al., 2017; Adnan et al., 2022), these models, while proficient with in-distribution data, may give overconfident or plainly incorrect outputs when faced with OOD samples (Wei et al., 2022).

We focus on detecting OOD inputs using likelihood-based deep generative models (DGMs), which aim to learn the density that generated the observed data. Maximum-likelihood operates by increasing model likelihoods on training data, and since probability densities must be normalized, one might reasonably expect lower likelihoods for OOD points. Likelihood-based DGMs, including variational autoencoders (Kingma & Welling, 2014; Rezende et al., 2014; Vahdat & Kautz, 2020), normalizing flows (NFs) (Dinh et al., 2016; Kingma & Dhariwal, 2018; Durkan et al., 2019), and diffusion models (Sohl-Dickstein et al., 2015; Ho et al., 2020; Song et al., 2021) have proven to be powerful DGMs that can render photo-realistic images. Given these successes, it seems reasonable to attempt OOD detection by thresholding the likelihood of a query datum under a trained model.

Surprisingly, likelihood-based DGMs trained on more complex datasets assign higher likelihoods to OOD datapoints from less complex datasets (Choi et al., 2018; Nalisnick et al., 2019a; Havtorn et al., 2021). This observation becomes even more puzzling in light of the fact that said DGMs are explicitly trained to assign high likelihoods to in-distribution data without having been exposed to OOD data, thus generating samples which are much closer to the former. In this work, we explore the following explanation for how both these observations can simultaneously be true:

> *OOD datapoints can be assigned higher likelihoods while not being generated*
> *if they belong to regions of low probability mass.*

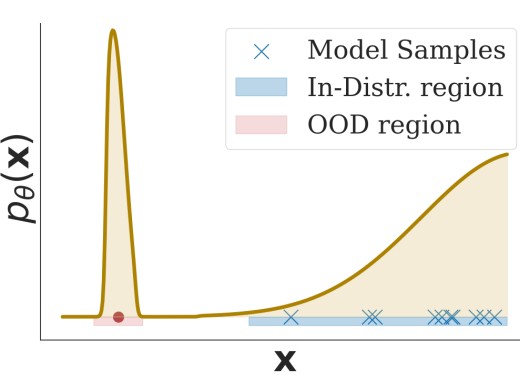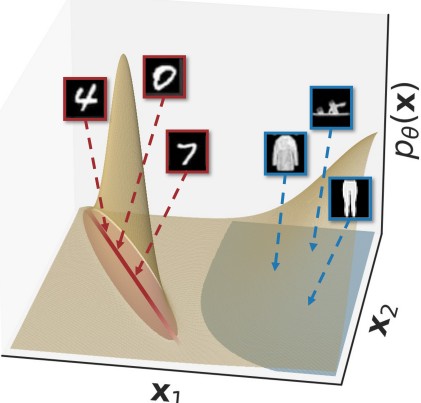

Figure 1: **Left:** A 1D density which is highly peaked in the OOD region (red) assigns high likelihood, but low probability mass to OOD data. **Right:** An analogous situation for a 2D density concentrated around a 1D OOD manifold (red line), illustrated with FMNIST as in-distribution and MNIST as OOD. The model density has become sharply peaked around the manifold of "less complex" data which has low intrinsic dimension, which is nonetheless assigned low probability mass as it has negligible volume.

Figure 1 illustrates that regions assigned high density by a model may integrate to very little probability mass when densities are sharply peaked. When OOD data is "simpler" in the sense that it concentrates on a manifold of lower dimension than in-distribution data, the phenomenon depicted in Figure 1 becomes completely consistent with empirical observations, but why might this happen in the first place? According to the manifold hypothesis, data often resides on lower-dimensional manifolds within a higher-dimensional space (Bengio et al., 2013; Pope et al., 2021; Brown et al., 2023). Consequently, a likelihood-based DGM aims to capture these manifolds by concentrating its density around them. Previous work has shown that DGMs have inductive biases that capture certain types of low-dimensional structure – structure that may also be present in OOD data – resulting in high assigned densities (Kirichenko et al., 2020; Schirrmeister et al., 2020). Our work is complementary to these past results: we provide a mathematical understanding of how the paradox can arise, which we leverage to devise a method to perform OOD detection using only a pre-trained density model.

**Contributions** ($i$) We develop an OOD detection method which classifies a datum as in-distribution only if it is assigned high likelihood by a DGM whose density is not sharply peaked around the datum, as measured by a large local intrinsic dimension (LID) estimate of the learned manifold; ($ii$) design an efficient LID estimator for our method, which requires only a pre-trained density model; ($iii$) we empirically verify the above explanation for the OOD paradox; and ($iv$) achieve state-of-the-art OOD detection performance among methods using the same NF backbone as us.

## 2 BACKGROUND

**Normalizing Flows** In this study, we target DGMs that produce a density model $p_\theta$, parameterized by $\theta$, which can be easily evaluated. Among them, NFs readily provide probability densities through the change of variables formula, making them suitable for studying pathologies in the likelihood function. A NF is a diffeomorphic mapping $f_\theta : \mathcal{Z} \to \mathcal{X}$ from a latent space $\mathcal{Z} = \mathbb{R}^d$ to data space $\mathcal{X} = \mathbb{R}^d$, which transforms a simple distribution $p_\mathcal{Z}$ on $\mathcal{Z}$, typically an isotropic Gaussian, into a complicated data distribution $p_\theta$ on $\mathcal{X}$. The change of variables formula allows one to evaluate the likelihood of a datum $\mathbf{x} \in \mathcal{X}$ as

$$\log p_\theta(\mathbf{x}) = \log p_\mathcal{Z}(\mathbf{z}) - \log |\det \boldsymbol{J}(\mathbf{z})|, \qquad (1)$$

where $\mathbf{z} = f_\theta^{-1}(\mathbf{x})$ and $\boldsymbol{J}(\mathbf{z})$ is the Jacobian of $f_\theta$ evaluated at $\mathbf{z}$. NFs are constructed in such a way that $\det \boldsymbol{J}(\mathbf{z})$ can be efficiently evaluated, and are trained through maximum-likelihood. After training, a sample is drawn from the latent distribution $\mathbf{z} \sim p_\mathcal{Z}$, and in turn, it is pushed through the mapping $f_\theta$ to produce a sample $f_\theta(\mathbf{z}) = \mathbf{x} \sim p_\theta$.

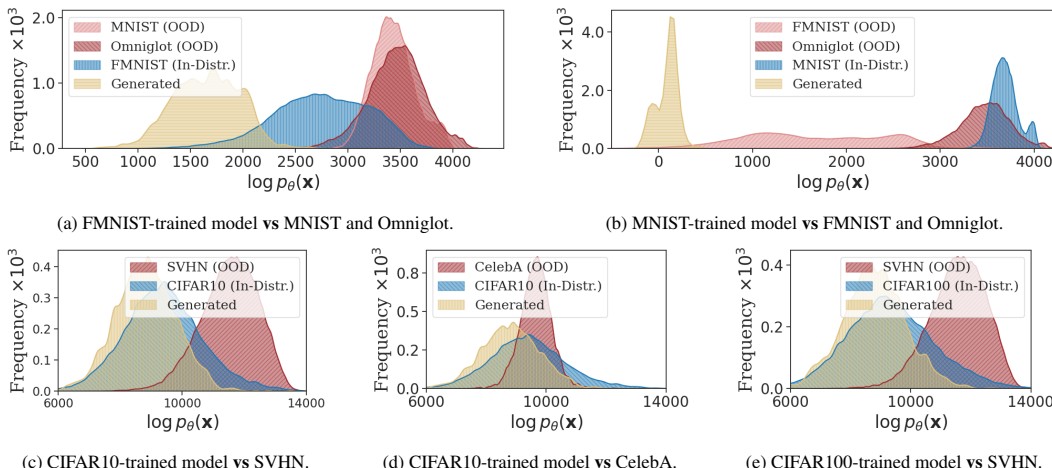

Figure 2: Overview of likelihood pathologies: (a) an FMNIST-trained model assigns higher likelihoods to MNIST and Omniglot data; (b) a model trained on MNIST shows notably lower likelihoods for its own generated samples compared to OOD data; (c-e) pathologies observed in some models trained on RGB datasets.

**Likelihood Pathologies in OOD Detection**   Choi et al. (2018) and Nalisnick et al. (2019a) first uncovered unintuitive behaviour that pervasively affects likelihood-based DGMs. For instance, models trained on relatively complex datasets like CIFAR10 (Krizhevsky & Hinton, 2009) and FMNIST (Xiao et al., 2017) often yield high likelihoods when tested on simpler ones like SVHN (Netzer et al., 2011) and MNIST (LeCun et al., 1998), respectively, despite the latter datasets not having been seen in the training process. While this issue is not exclusive to images (Ren et al., 2019), our experiments, shown in Figure 2, confirm these previous findings for NFs trained on image data. Additionally, we find that this pathological behaviour is not limited to these well-known cases, but extends to numerous dataset pairs and even to generated samples (see Appendix A for details).

**Local Intrinsic Dimension**   As previously discussed, natural data often lies on low-dimensional submanifolds of $\mathcal{X} = \mathbb{R}^d$, where $d$ is the *ambient* dimension of the data space. The *local intrinsic dimension* (LID) of a datapoint $\mathbf{x} \in \mathcal{X}$ with respect to these data submanifolds is the dimension of the submanifold that contains $\mathbf{x}$. For example, if the ambient space is $\mathbb{R}^2$ and the data manifold is the 1D unit circle $S$, then any point $\mathbf{x} \in S$ will have an intrinsic dimension of 1. Often, datasets lie on multiple non-overlapping submanifolds of different dimensionalities (Brown et al., 2023), in which case the LID will vary between datapoints.

Density models $p_\theta$ implicitly attempt to learn these manifolds by concentrating mass around them. As a consequence, even though they are defined on the full $d$-dimensional space $\mathcal{X}$, trained densities $p_\theta$ implicitly encode low-dimensional manifold structure, which in turn implies that $p_\theta$ contains LID information. When referring to an LID with respect to the manifolds implied by $p_\theta$, we will write $\mathrm{LID}_\theta(\mathbf{x})$. In the following, we will link LID to the sharpness of likelihood peaks, so it will be of interest to estimate $\mathrm{LID}_\theta(\mathbf{x}_0)$ for in- and out-of-distribution query points $\mathbf{x}_0$.

Various methods to estimate intrinsic dimension exist (Levina & Bickel, 2004; Johnsson et al., 2014; Facco et al., 2017; Bac et al., 2021). Unfortunately, most of these are inadequate for our purposes, either because they estimate global (i.e. averaged or aggregated) intrinsic dimension instead of $\mathrm{LID}_\theta(\mathbf{x}_0)$, or because they require observed data around $\mathbf{x}_0$ to produce the estimate. Since we want $\mathrm{LID}_\theta(\mathbf{x}_0)$ for OOD points $\mathbf{x}_0$, the latter methods would require access to samples from $p_\theta$ which fall in the OOD region, which are of course unavailable. The key to circumvent this issue is to use LID estimators based on densities rather than data.

Density-based LID estimators are enabled by a surprising result linking Gaussian convolutions and LID (Loaiza-Ganem et al., 2022; Tempczyk et al., 2022). Intuitively, adding high-dimensional but low-variance Gaussian noise corrupts $p_\theta$ more easily in lower-dimensional regions (see Figure 1 from Tempczyk et al. (2022)). Comparing $p_\theta$ convolved with noise for different noise levels allows one to infer LID from how quickly $p_\theta$ is corrupted as the noise increases. More formally, defining

the convolution between a pre-trained density $p_\theta$ and a Gaussian with log standard deviation $r$ as

$$\rho_r(\mathbf{x}) := [p_\theta(\cdot) * \mathcal{N}(\,\cdot\,; \mathbf{0}, e^{2r}\boldsymbol{I}_d)](\mathbf{x}) = \int p_\theta(\mathbf{x} - \mathbf{x}')\mathcal{N}(\mathbf{x}'; \mathbf{0}, e^{2r}\boldsymbol{I}_d)\mathrm{d}\mathbf{x}', \qquad (2)$$

Tempczyk et al. (2022) showed that under mild regularity conditions, for sufficiently negative $r$ (i.e. small standard deviation),

$$\log \rho_r(\mathbf{x}) = r(\mathrm{LID}_\theta(\mathbf{x}) - d) + \mathcal{O}(1). \qquad (3)$$

Equation 3 implies that, for sufficiently negative $r$ (corresponding to sufficiently small noise), the rate of change of $\log \rho_r(\mathbf{x}_0)$ with respect to $r$ can be used to estimate LID, since $\frac{\partial}{\partial r} \log \rho_r(\mathbf{x}_0) \approx \mathrm{LID}_\theta(\mathbf{x}_0) - d$. Tempczyk et al. (2022) leverage this observation to propose an estimator of $\mathrm{LID}_\theta(\mathbf{x}_0)$, which they call LIDL. To compute this estimate, they train one NF for each of several noise levels $r$ on in-distribution data convolved with $\mathcal{N}(\,\cdot\,; \mathbf{0}, e^{2r}\boldsymbol{I}_d)$. Then, for a given $\mathbf{x}_0$, they treat the log densities provided by the NFs as approximations of $\log \rho_r(\mathbf{x}_0)$, and they fit a simple linear regression to these values using $r$ as the covariate; the resulting slope is an estimate of $\mathrm{LID}_\theta(\mathbf{x}_0) - d$. This procedure is of course computationally intensive, as several NF models must be trained. We will later show how to obtain estimates of $\mathrm{LID}_\theta(\mathbf{x}_0)$ from a single pre-trained density model $p_\theta$.

## 3 RELATED WORK

A substantial amount of research into likelihood anomalies tries to explain the underlying causes for the OOD paradox. One particular line of research proposes probabilistic explanations: Choi et al. (2018) and Nalisnick et al. (2019b) put forth the "typical set" hypothesis, which has been contested in follow-up work. For example, Le Lan & Dinh (2021) argue that likelihood rankings not being invariant to data reparameterizations causes the paradox, whereas Caterini & Loaiza-Ganem (2022) claim it is the lower entropy of "simpler" distributions as compared to the higher entropy of more "complex" ones – which somewhat aligns with our work, although we use intrinsic dimension instead of entropy to quantify complexity.

Nonetheless, we significantly differ from these explanations in that they all assume, sometimes implicitly, that the supports of in- and out-of-distribution data overlap, and that the paradox is thus fundamentally unavoidable. We find it extremely plausible, for example, that the intersection between CIFAR10 and SVHN images is empty. In this sense, we are more in agreement with the work of Zhang et al. (2021) who blame model misalignment – i.e. $p_\theta$ failing to properly learn its target data-generating density – and highlight that perfect density models would not be subject to the OOD paradox if dataset supports do not overlap.

Previous work has also found an empirical cause of this model misalignment: Kirichenko et al. (2020) and Schirrmeister et al. (2020) show that the multi-scale convolutional architectures employed by NFs fixate on high-frequency local features and pixel-to-pixel correlations. When these features and correlations are strongly present in OOD data – which happens when it is "simpler" – OOD likelihoods become large. Although insightful, these studies do not provide an explanation for why OOD data is never generated. Our work is thus complementary to this research, as we show why the OOD paradox is mathematically possible, and use the resulting insights for OOD detection.

Another line of work aims to build DGMs which do not experience the aforementioned misalignment, sometimes at the cost of generation quality. For example, Grathwohl et al. (2020) and Liu et al. (2020) argue that the training procedure of energy-based models (EBMs) (Xie et al., 2016; Du & Mordatch, 2019) provides inductive biases which are useful for OOD detection, and Yoon et al. (2021) construct an EBM specifically designed for this task. Kirichenko et al. (2020) and Loaiza-Ganem et al. (2022) first embed data into semantically rich latent spaces, and then employ dense neural network architectures, thus minimizing susceptibility to local high-frequency features. We differ from these works in that they attempt to build DGMs that are better at likelihood-based OOD detection, whereas we only leverage a (potentially misaligned) pre-trained model. The understanding of the OOD paradox that we derive is nonetheless compatible with other DGMs.

Finally, other works use "outside help" or auxiliary models. Some methods assume access to an OOD dataset (Nalisnick et al., 2019b), require class labels (Görnitz et al., 2013; Ruff et al., 2020), or leverage an image compression algorithm (Serrà et al., 2020). Some other works, while fully unsupervised, require training an auxiliary model on distorted data (Ren et al., 2019) or on specific

data statistics (Morningstar et al., 2021). Once again, we differ from these works in that our focus is on gaining a deeper understanding of the OOD paradox, and using it for fully unsupervised OOD detection based only on a single pre-trained model.

## 4 METHOD

In this section, we provide an OOD detection algorithm that can be readily applied to any pre-trained NF, *even if it is misaligned* and exhibits undesired likelihood peaks on OOD data. In order to determine if a given query point $\mathbf{x}_0$ is in- or out-of-distribution, our method requires approximating $\log \rho_r(\mathbf{x}_0)$. While our main ideas are widely applicable to likelihood-based DGMs, we focus on NFs since, as we will shortly show, they allow for particularly tractable approximations. Intuitively, the fact that NFs never generate OOD samples indicates that they do contain the information needed to discern OOD from in-distribution samples, and we will show how to extract this information.

### 4.1 PROBABILITY MASS, SHARP LIKELIHOOD PEAKS, AND INTRINSIC DIMENSION

**Approximating Probability Mass**    Our explanation of the paradox involves OOD regions having sharply peaked likelihoods with negligible probability mass. It is thus natural to explore ways of computing the probability mass around $\mathbf{x}_0$ to determine if it is OOD or not. A sensible first attempt proposed by Grathwohl et al. (2020) is to consider the $\ell_2$ norm of the derivative of $\log p_\theta(\mathbf{x})$ with respect to $\mathbf{x}$ evaluated at $\mathbf{x}_0$, $\|\frac{\partial}{\partial \mathbf{x}} \log p_\theta(\mathbf{x}_0)\|_2$. Intuitively, a large norm would suggest a sharply peaked density and thus low probability mass; however, as they report and we confirm in Section 5, this produces unreliable results for NFs. A natural alternative would be to directly approximate the probability mass that the model assigns to a small neighbourhood of $\mathbf{x}_0$, such as an $\ell_2$ ball of radius $R$, $B_R(\mathbf{x_0}) := \{\mathbf{x} \in \mathcal{X} : \|\mathbf{x} - \mathbf{x}_0\|_2^2 \le R^2\}$:

$$\mathbb{P}_\theta\left(\mathbf{x} \in B_R(\mathbf{x}_0)\right) = \int_{B_R(\mathbf{x_0})} p_\theta(\mathbf{x})\mathrm{d}\mathbf{x} = \mathrm{vol}(B_R(\mathbf{0})) \cdot [p_\theta(\cdot) * \mathcal{U}(\,\cdot\,; B_R(\mathbf{0}))]\,(\mathbf{x}_0), \qquad (4)$$

where $\mathrm{vol}(B)$ denotes the $d$-dimensional Lebesgue measure (i.e. the volume) of $B$ and $\mathcal{U}(\,\cdot\,; B)$ the density of the uniform distribution on $B$. Since $\mathrm{vol}(B_R(\mathbf{0}))$ does not depend on $\mathbf{x}_0$, Equation 4 suggests approximating $[p_\theta(\cdot) * \mathcal{U}(\,\cdot\,; B_R(\mathbf{0}))](\mathbf{x}_0)$ for OOD detection. In initial experiments, we tried doing so through a Monte Carlo estimate, but found the estimator particularly unreliable (see Appendix B). More specifically, despite the true probability in Equation 4 being non-decreasing in $R$, its Monte Carlo estimate did not exhibit this property. To circumvent this issue, we leverage the standard and well-known result that in high dimensions, the uniform distribution on the ball is approximately Gaussian, $\mathcal{U}(\,\cdot\,; B_{e^r \sqrt{d}}(\mathbf{0})) \approx \mathcal{N}(\,\cdot\,; \mathbf{0}, e^{2r}\boldsymbol{I}_d)$.[1] This result suggests that, if we take $R = e^r \sqrt{d}$, we can approximate $[p_\theta(\cdot) * \mathcal{U}(\,\cdot\,; B_R(\mathbf{0}))](\mathbf{x}_0)$ using the density $\rho_r(\mathbf{x}_0)$ defined in Equation 2.

**Fast Convolutions of NFs with Gaussians**    In LIDL, estimating $\rho_r(\mathbf{x}_0)$ requires training a separate density model for each $r$ value of interest. This is computationally expensive and ill-suited for our purposes, since our goal is to use only $p_\theta$. Instead, we propose a way to leverage the properties of NFs to approximate $\rho_r(\mathbf{x}_0)$. Using a first order Taylor approximation of $f_\theta$ around $\mathbf{z}_0 = f_\theta^{-1}(\mathbf{x}_0)$, we approximate the NF as an affine function $f_\theta(\mathbf{z}) \approx \boldsymbol{J}_0(\mathbf{z} - \mathbf{z}_0) + \mathbf{x}_0$, where $\boldsymbol{J}_0$ is the Jacobian of $f_\theta$ evaluated at $\mathbf{z}_0$ and is tractable by design. Since $p_{\mathcal{Z}}$ is Gaussian and affine transformations of Gaussians remain Gaussian, we can thus approximate $p_\theta(\mathbf{x})$ as $\hat{p}_\theta(\mathbf{x}) := \mathcal{N}(\mathbf{x}; \mathbf{x}_0 - \boldsymbol{J}_0\mathbf{z}_0, \boldsymbol{J}_0\boldsymbol{J}_0^\top)$. While it might at first appear strange to approximate a function $f_\theta$ and a density $p_\theta$ that we can already evaluate, this approximation makes convolutions analytically tractable. By convolving $\hat{p}_\theta$ (instead of $p_\theta$) with a Gaussian, we can approximate $\rho_r(\mathbf{x}_0)$ from Equation 2 as

$$\hat{\rho}_r(\mathbf{x}_0) := \int \hat{p}_\theta(\mathbf{x}_0 - \mathbf{x})\mathcal{N}(\mathbf{x}; \mathbf{0}, e^{2r}\boldsymbol{I}_d)\mathrm{d}\mathbf{x} = \mathcal{N}(\mathbf{x}_0; \mathbf{x}_0 - \boldsymbol{J}_0\mathbf{z}_0, \boldsymbol{J}_0\boldsymbol{J}_0^\top + e^{2r}\boldsymbol{I}_d). \qquad (5)$$

---

[1]Readers unfamiliar with this can see Saremi & Hyvärinen (2019) for a related derivation in a machine learning context. Note that this derivation shows Gaussians are approximately uniform on the boundary of the ball, but another classic result is that, in high dimensions, the majority of the mass of the ball lies near its boundary (see e.g. Wegner (2021)), so that uniforms on the ball or its boundary are also approximately equal.

Importantly, since the Taylor expansion of $f_\theta$ is only accurate around $\mathbf{x}_0$, so is $\hat{p}_\theta$. In turn, $\hat{\rho}_r(\mathbf{x}_0)$ is an accurate approximation of $\rho_r(\mathbf{x}_0)$ only when $\mathcal{N}(\cdot; \mathbf{0}, e^{2r}\mathbf{I}_d)$ concentrates most of its mass around $\mathbf{0}$ – i.e. when $r$ is sufficiently negative – since this means that the contributions of $\hat{p}_\theta(\mathbf{x}_0 - \mathbf{x})$ to the integral in Equation 5 happen mostly when $\mathbf{x} \approx 0$. For a detailed analysis of the quality of this approximation, please refer to Appendix C.

**Identifying Sharp Likelihood Peaks** So far, we have argued that: $(i)$ for an appropriate $r$, $\rho_r(\mathbf{x}_0)$ is a sensible proxy for how much mass $p_\theta$ assigns around $\mathbf{x}_0$, and that $(ii)$ $\hat{\rho}_r(\mathbf{x}_0)$ is a good approximation of $\rho_r(\mathbf{x}_0)$ for negative enough values of $r$. However, these two arguments together do *not* necessarily imply that $\hat{\rho}_r(\mathbf{x}_0)$ estimates how much mass $p_\theta$ assigns around $\mathbf{x}_0$, since choices of $r$ high enough to capture the mass might not be negative enough for $\hat{\rho}_r(\mathbf{x}_0)$ to remain a close approximation of $\rho_r(\mathbf{x}_0)$. In Figure 3, which shows $\log \hat{\rho}_r$ for in- and out-of-distribution data as $r$ increases, some relevant patterns emerge.

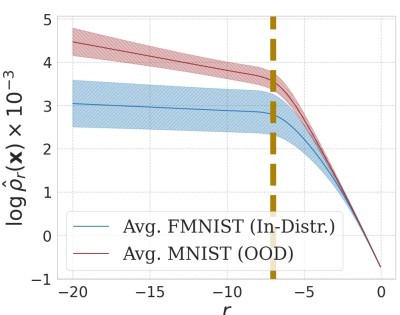

Figure 3: Comparison of mean values of $\hat{\rho}_r(\mathbf{x}_{\text{in}})$ and $\hat{\rho}_r(\mathbf{x}_{\text{ood}})$ for in- and out-of-distribution $\mathbf{x}_{\text{in}}$ and $\mathbf{x}_{\text{ood}}$, respectively. The shaded area shows one standard deviation. Here, in-distribution is FMNIST, and OOD is MNIST.

First, we can see a marked change of behaviour around $r \approx -7$. Since $\rho_r(\mathbf{x}_0) \approx \hat{\rho}_r(\mathbf{x}_0)$ for negative enough values of $r$, we believe that the consistent behaviour before $r \approx -7$ agrees with our approximation remaining accurate until this threshold. However, it is unclear if the behaviour after $r \approx -7$ changes due to an underlying change in $\rho_r$ or because of a decline in approximation quality. We thus find it prudent to only trust $\hat{\rho}_r$ for negative enough values of $r$ below this threshold. Second, before $r \approx -7$, OOD values of $\hat{\rho}_r$ remain larger, which is consistent with $p_\theta$ also being larger for OOD data and with $r$ being too negative to appropriately quantify the probability mass around $\mathbf{x}_0$.

Fortunately, even though the values of $r$ for which $\hat{\rho}_r(\mathbf{x}_0)$ is a good approximation of $\rho_r(\mathbf{x}_0)$ are too negative for $\rho_r(\mathbf{x}_0)$ to remain a useful proxy for probability mass, $\hat{\rho}_r(\mathbf{x}_0)$ nonetheless contains relevant information. In the example shown in Figure 3, it is clear that $\log \hat{\rho}_r(\mathbf{x}_0)$ decreases faster for pathological OOD data than it does for in-distribution data, including in the regime where $r$ is negative enough for us to trust our approximations. Thus, instead of attempting to estimate $\log \rho_r(\mathbf{x}_0)$ for relatively less negative values of $r$, we estimate its rate of change, $\frac{\partial}{\partial r} \log \rho_r(\mathbf{x}_0)$, for very negative values of $r$. Intuitively, the rate of change that we consider, $\frac{\partial}{\partial r} \log \hat{\rho}_r(\mathbf{x}_0)$, can be understood as a measure of how sharply peaked $p_\theta$ is around $\mathbf{x}_0$.

Finally, we make a crucial observation: $\frac{\partial}{\partial r} \log \rho_r(\mathbf{x}_0)$ is *exactly* what LIDL aims to estimate from Equation 3, i.e. $\text{LID}_\theta(\mathbf{x}_0) - d$. Importantly, we now have a well-grounded understanding that $\text{LID}_\theta(\mathbf{x}_0) \approx \frac{\partial}{\partial r} \log \hat{\rho}_r(\mathbf{x}_0) + d$ provides a measure of how sharply peaked $p_\theta$ is around $\mathbf{x}_0$, thus making LID a perfect candidate for OOD detection.

## 4.2 Putting it All Together: Dual Threshold OOD Detection

In summary, three mutually exclusive circumstances can arise: $(i)$ $p_\theta(\mathbf{x}_0)$ is very small, in which case we can infer that $\mathbf{x}_0$ does not lie on the manifolds learned by $p_\theta$, so we should flag $\mathbf{x}_0$ as OOD. Non-pathological situations where the model $p_\theta$ actually assigns very low likelihoods to OOD data will of course be detected in this case. $(ii)$ $p_\theta(\mathbf{x}_0)$ is large, and $\frac{\partial}{\partial r} \log \hat{\rho}_r(\mathbf{x}_0) \approx \text{LID}_\theta(\mathbf{x}_0) - d$ is very negative, i.e. $\text{LID}_\theta(\mathbf{x}_0)$ is very small. Here, the likelihood is high but sharply peaked, suggesting that $p_\theta$ places a negligible amount of mass

**Algorithm 1** Dual threshold OOD detection, returns `True` if $\mathbf{x}_0$ is deemed OOD, and `False` if it is deemed in-distribution.

---
**Require:** $\mathbf{x}_0, p_\theta, r, \psi_\mathcal{L}, \psi_{\text{LID}}$
1: **if** $\log p_\theta(\mathbf{x}_0) < \psi_\mathcal{L}$ **then**
2:    **return** `True`          $\triangleright$ case $(i)$
3: **if** $\frac{\partial}{\partial r} \log \hat{\rho}_r(\mathbf{x}_0) < \psi_{\text{LID}}$ **then**
4:    **return** `True`          $\triangleright$ case $(ii)$
5: **return** `False`          $\triangleright$ case $(iii)$

---

around $\mathbf{x}_0$. In this case, we should also flag $\mathbf{x}_0$ as OOD. As previously mentioned, we see this case as caused by misalignment of $p_\theta$, yet our LID estimate still allows us to detect this pathological situation. $(iii)$ $p_\theta(\mathbf{x}_0)$ is large, and $\frac{\partial}{\partial r}\log\hat{\rho}_r(\mathbf{x}_0) \approx \text{LID}_\theta(\mathbf{x}_0) - d$ is not very negative, i.e. $\text{LID}_\theta(\mathbf{x}_0)$ is also large. In this case the likelihood is large but not sharply peaked, suggesting that $p_\theta$ places a relatively large amount of mass around $\mathbf{x}_0$, so we can tag $\mathbf{x}_0$ as in-distribution.

Putting these three possibilities together, we propose a simple OOD detection method in Algorithm 1 which uses two thresholds, $\psi_\mathcal{L}$ and $\psi_{\text{LID}}$, for $\log p_\theta(\mathbf{x}_0)$ and $\frac{\partial}{\partial r}\log\hat{\rho}_r(\mathbf{x}_0)$, respectively, to decide which case $\mathbf{x}_0$ belongs to. In Appendix D.1 we detail how we select all the hyperparameters ($r$, $\psi_\mathcal{L}$, and $\psi_{\text{LID}}$), but make the important observation that we set them in a fully unsupervised way, i.e. *without* relying on OOD data, using only in-distribution training data and the given model $p_\theta$.

## 5 EXPERIMENTS

**Setup**  For a thorough examination of likelihood pathologies, we compared datasets within two classes: $(i)$ $28 \times 28$ greyscale images, including FMNIST, MNIST, Omniglot (Lake et al., 2015), and EMNIST (Cohen et al., 2017); and $(ii)$ RGB images resized to $32 \times 32 \times 3$, comprising SVHN, CIFAR10 and CIFAR100 (Krizhevsky & Hinton, 2009), Tiny ImageNet (Le & Yang, 2015), and a simplified, cropped version of CelebA (Kist, 2021). While we kept the most popular OOD detection tasks in the main text, due to space constraints our extensive experiments across all of the OOD detection tasks along with details of our setup for reproducibility are available in Appendix D.2. Interestingly, we did not see the exploding inverses identified by Behrmann et al. (2021) in our flow models. We postulate this could be due to a different hyperparameter configuration than theirs; see Figure 10 in Appendix D.1. Our anonymized code, which will be made publicly available, can be found here `https://anonymous.4open.science/r/LID-OOD-1E25`.

**Notation**  Throughout this section, we will use the suffixes "-train", "-test", and "-gen" when talking about datasets to specify if we are referring to the train set, test set, or generated samples, respectively. For datasets $A$ and $B$, we write "$A$ (vs) $B$" to refer to the OOD detection task that aims to distinguish $A$-test from $B$-test using the model $p_\theta$ pre-trained on $A$-train. When we write "$A$-gen (vs) $B$", $A$-test is replaced by $A$-gen, but $p_\theta$ is still pre-trained on $A$-train. Ideally, for a model trained on $A$-train, both $A$-gen and $A$-test should align. However, if the model excels in the "$A$-gen (vs) $B$" task but falters in "$A$ (vs) $B$", it indicates poor model fit. By making these comparisons, we can pinpoint scenarios where improved generative models are required.

**Evaluation**  For the single threshold baselines, we employ the area under the curve (AUC) of receiver operator curves (ROC), a common metric for classifier performance, to evaluate them on OOD detection tasks. Notably, dual threshold classifiers do not produce a traditional ROC; however, they give rise to a receiver operator graph (ROG). Each dual threshold, represented by $(\psi_\mathcal{L}, \psi_{\text{LID}})$, maps to a point on the ROG, indicating the true-positive (TP) and false-positive (FP) rates of the corresponding classifier. The ROG's convex hull (including the axes from 0 to 1) acts as a ROC surrogate, allowing for comparisons against the AUCs of single threshold baselines.

**Together, Likelihoods and LID Isolate OOD Regions**  In our first set of experiments, we compute log-likelihoods and LID estimates for each datapoint. Results are shown in Figure 4. When considered separately, the likelihoods or LIDs of in- and out-of-distribution data can overlap, as depicted by the likelihood marginals on the left and right, and the LID marginals on the bottom. In contrast, the scatterplots using both likelihoods and LIDs show a clear separation between in-distribution and OOD datapoints. Furthermore, the "directions" predicted by our method are correct: in the pathological case, FMNIST (vs) MNIST (left), we clearly see that while OOD points have higher likelihoods, they also have lower LIDs; whereas in the non-pathological case, MNIST vs FMNIST (right), likelihoods are lower for OOD data. These results highlight not only the importance of using LID estimates for OOD detection, but also that of combining them with likelihoods. For more comparisons depicting the same scatterplots across various datasets, please refer to Appendix D.3.

**Visualizing the Benefits of Dual Thresholding**  The separation of OOD and in-distribution data shown in the scatterplots in Figure 4 confirms that likelihood/LID pairs contain the needed information for OOD detection. However, it remains to show that Algorithm 1 succeeds at this task (recall

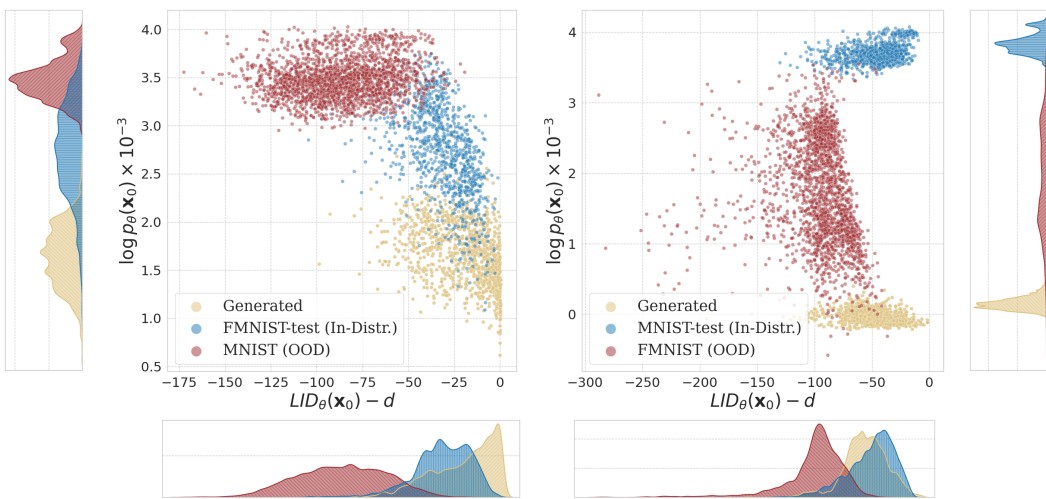

Figure 4: LID estimates and likelihood values: FMNIST (vs) MNIST and FMNIST-gen (vs) MNIST (**left scatterplot**); and MNIST (vs) FMNIST and MNIST-gen (vs) FMNIST (**right scatterplot**). Marginal distributions of likelihoods and LIDs are shown on the sides and bottom, respectively.

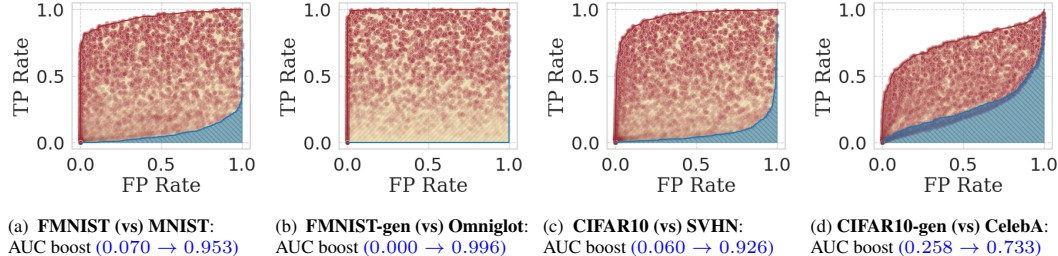

(a) **FMNIST (vs) MNIST**: AUC boost $(0.070 \rightarrow 0.953)$

(b) **FMNIST-gen (vs) Omniglot**: AUC boost $(0.000 \rightarrow 0.996)$

(c) **CIFAR10 (vs) SVHN**: AUC boost $(0.060 \rightarrow 0.926)$

(d) **CIFAR10-gen (vs) CelebA**: AUC boost $(0.258 \rightarrow 0.733)$

Figure 5: ROG visualizations for select pathological OOD tasks. The red dots and the yellow area correspond to the ROG graph, and the ROC for our dual thresholding method, respectively; the blue areas represent ROCs for single threshold likelihood-based classifiers.

that we cannot simply train a classifier to differentiate between red and blue points in Figure 4 since the red OOD points are unavailable when designing the OOD detector). Figure 5 provides a visual comparison showcasing the ROGs from our dual thresholding technique versus the ROCs obtained by single threshold classifiers obtained by using only likelihoods. These results not only show a dramatic boost in AUC-ROC performance across four different pathological scenarios – once again highlighting the relevance of combining likelihoods with LIDs for OOD detection – but they also showcase that our dual threshold method successfully carries out this combination. For further experiments showing why dual thresholding is necessary and how it can boost ROC-AUC across all the OOD detection tasks we consider, please refer to Appendix D.4 and Appendix D.5.

**Quantitative Comparisons** In the top part of Table 1, we compare our method against several baselines, all of which are evaluated using the exact same NF as our method. These baselines are: $(i)$ naïvely labeling large likelihoods $\log p_\theta(\mathbf{x}_0)$ as in-distribution, which as previously mentioned, strongly fails at identifying "simpler" distributions as OOD when trained on "complex" datasets (e.g. FMNIST (vs) MNIST, FMIST (vs) Omniglot, and CIFAR10 (vs) SVHN). $(ii)$ Using $\|\frac{\partial}{\partial \mathbf{x}} \log p_\theta(\mathbf{x}_0)\|_2$, as proposed by Grathwohl et al. (2020), which performs very poorly across tasks, except at CIFAR10 (vs) SVHN. $(iii)$ The complexity correction method of Serrà et al. (2020), which uses image compression information to adjust the inflated likelihood observed in OOD datapoints. Despite this comparison being unfair in that the baseline is allowed access not only to the NF, but to image compression algorithms as well, we beat it across all tasks except for FMNIST (vs) Omniglot. $(iv)$ The likelihood ratios approach of Ren et al. (2019), which employs an auxiliary likelihood-based reference model to compute this ratio. This comparison is once again unfair, as the baseline has access to an additional model that we do not, and yet we uniformly beat it across tasks.

Table 1: ROC-AUC (higher is better). The top part of the table contains NF-based approaches, whereas the last row shows an EBM-based one. **Notation**: [*] tasks where likelihoods alone do not exhibit pathological behaviour, [‡] methods that employ external information or auxiliary models. For the NF methods, we bold the best performing model, and the EBM model is bolded when it surpasses all others.

| Trained on | MNIST [*] | | FMNIST | | CIFAR10 | | SVHN [*] | |
|---|---|---|---|---|---|---|---|---|
| OOD Dataset | FMNIST | Omniglot | MNIST | Omniglot | SVHN | CelebA | CIFAR10 | CelebA |
| Naïve Likelihood $\log p_\theta(\mathbf{x}_0)$ | **1.000** | 0.807 | 0.069 | 0.088 | 0.084 | 0.386 | **0.987** | **0.995** |
| $\|\frac{\partial}{\partial \mathbf{x}} \log p_\theta(\mathbf{x}_0)\|_2$ | 0.156 | 0.444 | 0.516 | 0.538 | 0.722 | 0.433 | 0.200 | 0.080 |
| Complexity Correction[‡] | 0.945 | 0.852 | 0.939 | **0.935** | 0.835 | 0.479 | 0.771 | 0.639 |
| Likelihood Ratios[‡] | 0.944 | 0.722 | 0.666 | 0.639 | 0.299 | 0.396 | 0.302 | 0.099 |
| Dual Threshold (Ours) | **1.000** | **0.869** | **0.953** | 0.862 | **0.926** | **0.653** | **0.987** | **0.995** |
| NAE | **1.000** | **0.994** | **0.995** | **0.976** | **0.919** | **0.887** | 0.948 | 0.965 |

Besides the strong empirical performance of our method, other aspects of the top part of Table 1 warrant attention. Both the complexity correction and likelihood ratio baselines lose performance over naïvely using likelihoods on non-pathological tasks, i.e. when models are trained on relatively "simple" data like MNIST or SVHN. Since likelihoods perform well at these tasks, they are often considered "easy" and thus omitted from comparisons. The fact that complexity correction and likelihood ratios struggle at these tasks is a novel finding that suggests these methods "overfit" to the pathological tasks, and highlights the underlying difficulty of unsupervised OOD detection. See Appendix E for additional results and discussions. Overall, we believe it is remarkable that our dual threshold method so conclusively outperforms these baselines at both pathological and non-pathological tasks, despite them having access to additional help. We see these results as strong evidence supporting the understanding that we derived about the OOD paradox and its connection to local intrinsic dimension.

The last row of Table 1 shows the performance of normalized autoencoders (NAEs) (Yoon et al., 2021). NAEs are EBMs specially tailored for OOD detection at the cost of generation quality, but to the best of our knowledge achieve state-of-the-art performance on fully unsupervised, likelihood-based OOD detection. Once again, we believe that the empirical results of our dual threshold method are remarkable: we achieve similar performance to NAEs on most tasks, even outperforming them on three, despite using a general purpose model $p_\theta$, not one explicitly designed for OOD detection.

## 6 CONCLUSIONS, LIMITATIONS, AND FUTURE WORK

In this paper we studied the OOD detection paradox, where likelihood-based DGMs assign high likelihoods to OOD points from "simpler" datasets, but do not generate them. We proposed an explanation of how the paradox can arise as a consequence of sharply peaked densities in OOD regions, thus assigning low probability mass to them. We connected this explanation to LID and proposed an efficient estimator of LID which we leveraged for our dual threshold OOD detection method, thus beating every method that used the same underlying NF model. While our ideas are widely applicable to any density model, the current incarnation of our method is limited in that it only applies to NFs, as estimating LID is more tractable for these models. Extending our methodology to other likelihood-based DGMs which map a Gaussian latent space through a readily available decoder, such as variational autoencoders or others (Brehmer & Cranmer, 2020; Caterini et al., 2021; Ross & Cresswell, 2021), can likely also work well with a similar Taylor expansion of the decoder. Nonetheless, we see extending our method to diffusion models, which achieve state-of-the-art performance for image synthesis, and to EBMs, which achieve state-of-the-art likelihood-based OOD detection, to be particularly promising directions for future work.

**Reproducibility Statement** To facilitate the reproducibility of our experiments we have provided a link to our anonymized codebase. It contains instructions on how to build an environment, and on how to run the code to repeat our experiments. Our methods are described in Section 4 and we have included pseudocode for the algorithms we propose in Algorithm 1. All necessary details of the experimental setup are given in Section 5 and Appendix D. All datasets we used are freely available for download from the cited sources and are used in accordance with any applicable licenses.

**Ethics Statement**   We do not foresee any ethics concerns with the present research. The overarching topic, OOD detection, is widely used as a method to improve the reliability of machine learning models in critical settings. Our goal is to theoretically explain and empirically rectify issues that arise when using OOD detection with deep generative models, and does not promote the use of generative models for harmful applications.

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

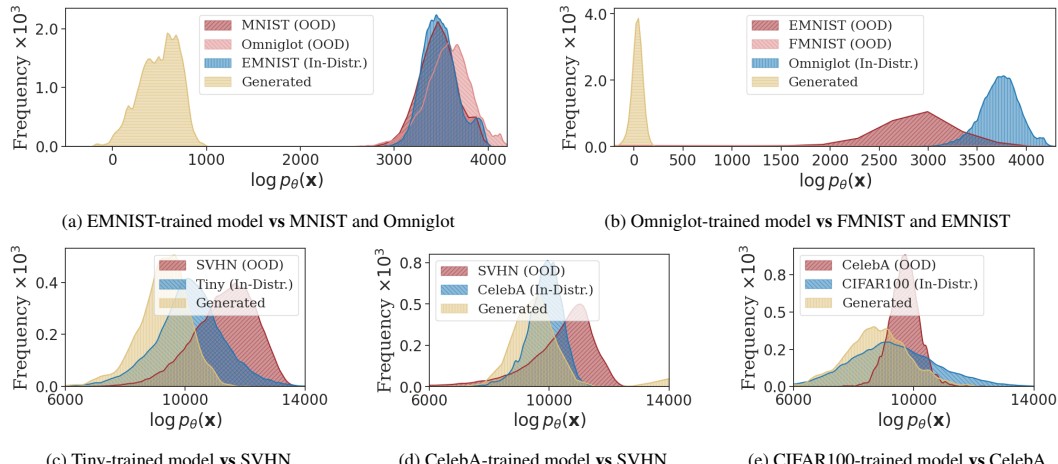

Figure 6: Extra Likelihood Pathologies Overview: (a) A model trained on the EMNIST dataset yields marginally lower likelihoods for its in-distribution data than for OOD data from Omniglot and strikingly low likelihoods on its generated samples. (b) An Omniglot-trained model displays low likelihoods for its own generated samples. (c-e) Additional pathologies among our RGB dataset pairs: Tiny ImageNet (vs) SVHN, CelebA (vs) SVHN, and CIFAR100 (vs) CelebA.

## A DIAGNOSING PATHOLOGIES IN FLOW MODELS

In this section we list the full extent of the pathologies we identified in our experiments. The first class is the standard one, in which NFs assign equal or higher likelihoods to out-of-distribution data than the distribution of the data they were trained on. In addition to Figure 2, which shows that FMNIST (vs) MNIST, CIFAR-10 (vs) SVHN, CIFAR10 (vs) CelebA, and CIFAR100 (vs) SVHN are pathological, Figure 6 depicts pathological behaviour for EMNIST (vs) MNIST, EMNIST (vs) Omniglot, Tiny (vs) SVHN, CelebA (vs) SVHN, and CIFAR100 (vs) CelebA.

Moreover, since models may not be perfectly fit, the likelihoods obtained for generated samples may not align with those of the test split of the in-distribution data. To demonstrate this phenomenon, we visualize the likelihoods of generated samples in Figure 2 and Figure 6. Notably, these likelihoods are almost always smaller than the in-distribution samples; this is a new, second class of pathologies. This difference is especially evident in greyscale models like FMNIST, MNIST, EMNIST, and Omniglot. Here, generated samples consistently show lower likelihoods than both in-distribution *and OOD data* points, as seen in Figure 2a, Figure 2b, Figure 6a, and Figure 6b respectively. Some of these cases, such as MNIST- and Omniglot-trained models were previously thought to be non-pathological (Nalisnick et al., 2019a). This adds to the unexplained phenomena of likelihood-based DGMs for OOD detection.

# B  COMPUTATIONAL INTRACTABILITY OF PROBABILITY MASSES

As mentioned in the main text, one could try to approximate the probability in Equation 4. A natural solution would be a Monte Carlo estimator:

$$\mathbb{P}_\theta(\mathbf{x} \in B_R(\mathbf{x}_0)) = \text{vol}(B_R(\mathbf{x}_0)) \cdot \mathbb{E}_{\mathbf{x}\sim\mathcal{U}(\,\cdot\,;B_R(\mathbf{x}_0))}\left[p_\theta(\mathbf{x})\right] \approx \text{vol}(B_R(\mathbf{x}_0)) \sum_{i=1}^{n} p_\theta(\mathbf{x}^{(i)}), \quad (6)$$

where $\mathbf{x}^{(i)} \sim \mathcal{U}(\,\cdot\,;B_R)$. Note that $\text{vol}(B_R(\mathbf{x}_0))$ admits a closed-form formula. Naïvely computing the above estimator is numerically unstable, but at the cost of some bias, $\log \mathbb{P}_\theta(\mathbf{x} \in B_R(\mathbf{x}_0))$ can be approximated instead by using the `logsumexp` function, since we can evaluate $\log p_\theta(\mathbf{x})$. In Figure 7 we show these estimates for a model trained on FMNIST over a range of small radii. Unlike $\log \mathbb{P}_\theta(\mathbf{x} \in B_R(\mathbf{x}_0))$, the estimated values are *not* non-decreasing in $R$, both for in- and out-of-distribution data: this highlights that these estimates are completely unreliable. While we believe Markov Chain Monte Carlo methods are a promising avenue towards more accurately approximating these probabilities, we leave such explorations for future work.

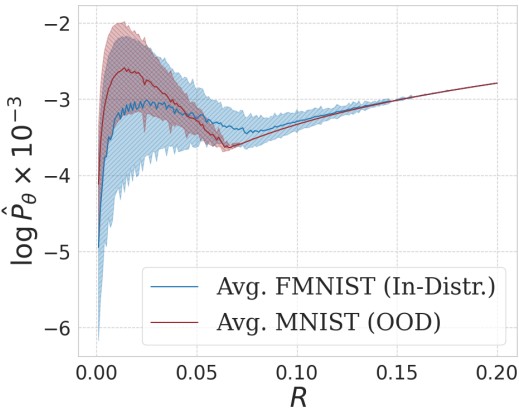

Figure 7: The trend of the average estimated $\log \mathbb{P}_\theta(\mathbf{x} \in B_R(\mathbf{x}_0))$ for MNIST and FMNIST on an FMNIST-trained NF.

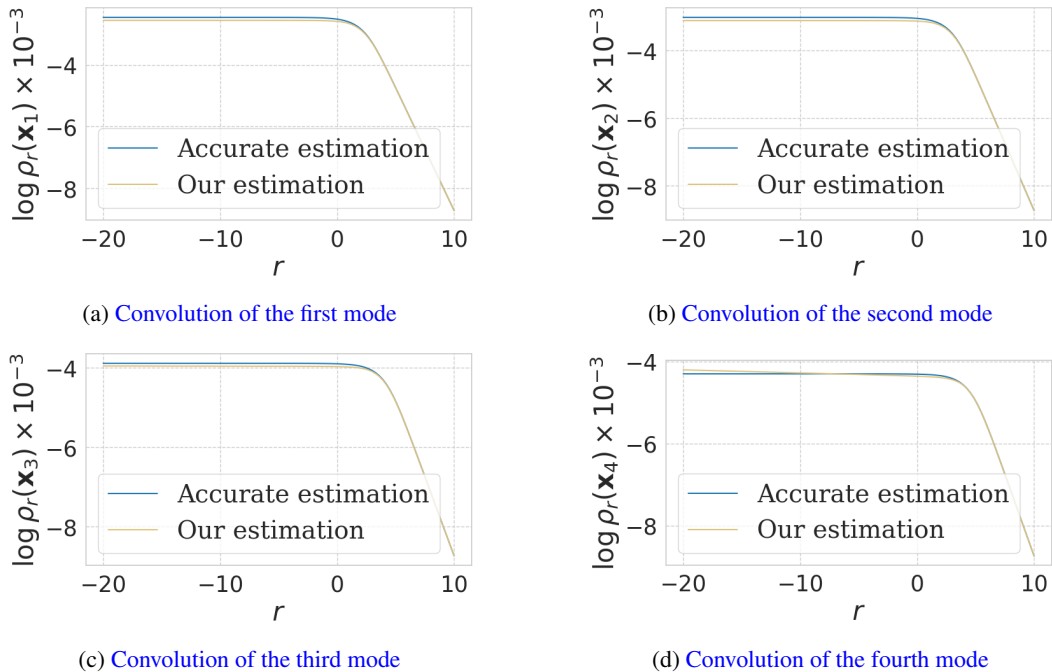

(a) Convolution of the first mode

(b) Convolution of the second mode

(c) Convolution of the third mode

(d) Convolution of the fourth mode

Figure 8: The estimated convolution $\log \rho_r(\mathbf{x}_i)$ for different modes of a mixture of Gaussians.

## C ON THE QUALITY OF THE LINEAR APPROXIMATION

In this section, we assess the accuracy of the approximation $\hat{\rho}_r$, as introduced in subsection 4.1.

### C.1 EMPIRICAL RESULTS

To empirically evaluate our approximation $\log \hat{\rho}_r$, we experiment on a synthetic dataset. Our experiment shows that $\log \hat{\rho}_r$ accurately represents $\log \rho_r$, the logarithm of a Gaussian convolved with the model's density, on this dataset.

First, we generated an $800$-dimensional dataset from a mixture of four Gaussians with different means and covariances. After training an RQ-NSF on $70,000$ samples of this distribution, we obtain a model $p_\theta$ that fits this mixture of Gaussians well. The modes of the distribution are set far apart and their componentwise covariance matrices are set so that there is minimal overlap between them; this setting allows us to derive an analytical expression to accurately approximate the density around each mode as follows. If $c_i$ is the probability of a sample belonging to the $i$th component with $\sum_i c_i = 1$, and $\mathbf{x}_i$ and $\mathbf{A}_i$ represent the mean and covariance matrix of the $i$th component, respectively, then the density at each mode can be approximated by ignoring the other components:

$$\log p_\theta(\mathbf{x}_i) \approx \log c_i \cdot \mathcal{N}(\mathbf{x}_i; \mathbf{x}_i, \mathbf{A}_i) \Rightarrow \log \rho_r(\mathbf{x}_i) \approx \log \frac{1}{\sqrt{(2\pi)^d \det(\mathbf{A}_i + e^{2r}\mathbf{I})}} \tag{7}$$

Therefore, comparing our estimator inspired by the Taylor expansion with the value obtained above provides us with a way to validate the quality of our estimator for $\log \rho_r$. Figure 8 (a-d) shows the estimated $\log \hat{\rho}_r$ for every mode vis-à-vis the accurate value obtained analytically from the above equation as $r$ increases. This comparison confirms that these estimators can be accurate approximations, even for high dimensional data.

### C.2 THEORETICAL BOUNDS

In addition to the empirical analysis, we mathematically evaluate the accuracy of our approximation by establishing error bounds. To advance our discussion, we first introduce two key mathematical operators essential for deriving these bounds.

**Definition C.1.** For positive scalars $\sigma$ and $\delta$, the *constrained Gaussian convolution operator* $\phi_\sigma^\delta$ : $\mathcal{F} \to \mathcal{F}$ on the family of smooth functions $\mathcal{F}$ mapping from $\mathbb{R}^d$ to $\mathbb{R}^+$ takes in an arbitrary function $h$ and outputs $g$ as follows:

$$\phi_\sigma^\delta(h) = g \text{ s.t. } g(\mathbf{x}) := \int_{B_{\sigma\delta}(\mathbf{x})} h(\mathbf{x} - \mathbf{u}) \cdot \mathcal{N}(\mathbf{u}; \mathbf{x}, \sigma^2 \cdot \mathbf{I}) \cdot d\mathbf{u}. \tag{8}$$

Here, the integral is on the $\ell_2$ ball of radius $\sigma \cdot \delta$. The *unconstrained Gaussian convolution operator* is also defined similarly with the exception that the integration is over the complement of the ball:

$$\bar{\phi}_\sigma^\delta(h) = g \text{ s.t. } g(\mathbf{x}) := \int_{\mathbb{R}^d \setminus B_{\sigma\delta}(\mathbf{x})} h(\mathbf{x} - \mathbf{u}) \cdot \mathcal{N}(\mathbf{u}; \mathbf{x}, \sigma^2 \cdot \mathbf{I}) \cdot d\mathbf{u}. \tag{9}$$

Note that $\phi_\sigma^\delta + \bar{\phi}_\sigma^\delta$ results in the normal convolution operator, and when the input of the operator is the density function $p_\theta$, we have that $\rho_r(\mathbf{x}_0) = \phi_\sigma^\delta(p_\theta)(\mathbf{x}_0) + \bar{\phi}_\sigma^\delta(p_\theta)(\mathbf{x}_0)$ for $\sigma = e^r$ and any $\delta$. Now we will provide an upper bound for the unconstrained convolution operator which will help us provide a global error margin for $\hat{\rho}_r(\mathbf{x}_0)$:

**Lemma C.1.** *Given a bounded function $h : \mathbb{R}^d \to \mathbb{R}^+$ where $M := \sup_{\mathbf{x} \in \mathbb{R}^d}(h(\mathbf{x}))$, we have that*

$$\bar{\phi}_\sigma^\delta(h)(\mathbf{x}_0) \le M \cdot \left[ (\delta/d)e^{1-(\delta/d)} \right]^{d/2}, \tag{10}$$

*when $\delta > d$.*

*Proof.*

$$\bar{\phi}_\sigma^\delta(h)(\mathbf{x}_0) = \int_{\mathbb{R}^d \setminus B_{\sigma\delta}(\mathbf{x}_0)} h(\mathbf{x}_0 - \mathbf{u}) \cdot \mathcal{N}(\mathbf{u}; \mathbf{x}_0, \sigma^2 \cdot \mathbf{I}) \cdot d\mathbf{u}$$

$$\le \int_{\mathbb{R}^d \setminus B_\delta(\mathbf{x}_0)} M \cdot \mathcal{N}(\mathbf{v}; \mathbf{0}, \mathbf{I}) \cdot d\mathbf{v} \qquad \text{Change of variables } \mathbf{v} := (\mathbf{u} - \mathbf{x}_0) \cdot \sigma^{-1}$$

$$= M \cdot P(\chi_d^2 > \delta^2) \le M \cdot \left[ (\delta/d)e^{1-(\delta/d)} \right]^{d/2},$$

where $\chi_d^2$ denotes a Chi-squared random variable with $d$ degrees of freedom. The last inequality is a well-known Chernoff bound on the survival function of the Chi-squared distribution. $\square$

Given an NF with a diffeomorphic mapping $f_\theta$, a high-quality local approximation to $f_\theta$ around $\mathbf{x}_0$ would result in a high-quality approximation of $p_\theta$ via the change of variables formula. Now we will present error bounds for our core estimator $\hat{\rho}_r(\mathbf{x}_0)$.

**Theorem C.2.** *Assume $R$ specifies a region in which an approximation $\hat{p}_\theta$ is accurate up to a small error margin: $\forall \mathbf{x} \in B_R(\mathbf{x}_0) : |\hat{p}_\theta(\mathbf{x}) - p_\theta(\mathbf{x})| \le e(R)$. If $M' := \sup_{\mathbf{x} \in supp(p_\theta)}(|p_\theta(\mathbf{x}) - \hat{p}_\theta(\mathbf{x})|)$ is finite and $e^r = \sigma < R/d$, then the total error between $\rho_r(\mathbf{x}_0)$ and $\hat{\rho}_r(\mathbf{x}_0)$ is bounded as follows:*

$$|\rho_r(\mathbf{x}_0) - \hat{\rho}_r(\mathbf{x}_0)| \le e(R) + M' \cdot \left[ \frac{R}{d\sigma} e^{1 - \frac{R}{d\sigma}} \right]^{d/2} \tag{11}$$

*Proof.*

$$|\rho_r(\mathbf{x}_0) - \hat{\rho}_r(\mathbf{x}_0)| \le \phi_\sigma^{R/\sigma}(|p_\theta - \hat{p}_\theta|)(\mathbf{x}_0) + \bar{\phi}_\sigma^{R/\sigma}(|p_\theta - \hat{p}_\theta|)(\mathbf{x}_0)$$

$$\le \int_{B_R(\mathbf{x}_0)} |p_\theta(\mathbf{x}_0 - \mathbf{u}) - \hat{p}_\theta(\mathbf{x}_0 - \mathbf{u})| \cdot \mathcal{N}(\mathbf{u}; \mathbf{x}_0, \sigma^2 \cdot \mathbf{I}) \cdot d\mathbf{u} + M' \cdot \left[ (R/d\sigma)e^{1-(R/d\sigma)} \right]^{d/2}$$

$$\le e(R) + M' \cdot \left[ (R/d\sigma)e^{1-(R/d\sigma)} \right]^{d/2}$$

$\square$

Therefore, for a sufficiently small $\sigma$ (translating to a sufficiently negative $r$) the total error bound of $\hat{\rho}_r$ is as small as the bound $e(R)$ obtained from the linearization. This analysis demonstrates that even though $\rho_r(\mathbf{x}_0)$ concerns taking a convolution over the entire support of $p_\theta$, a high-quality local approximation of the density directly yields a high-quality approximation $\hat{\rho}_r(\mathbf{x}_0)$.

# D EXPERIMENTAL DETAILS AND ADDITIONAL EXPERIMENTS

## D.1 DETAILS OF LID ESTIMATION

As mentioned in Section 4, our dual threshold method requires setting hyperparmeters. Here we describe how to set $r$. Naïvely, we could pick an arbitrary, negative enough value, but we found this does not work well across datasets as the results can be sensitive to the choice of $r$. Figure 9 shows the issue: setting the parameter this way will not properly detect the lower LID of SVHN with respect to a model $p_\theta$ trained on CIFAR10. A way to intuitively understand why the choice of $r$ matters is that it cor-

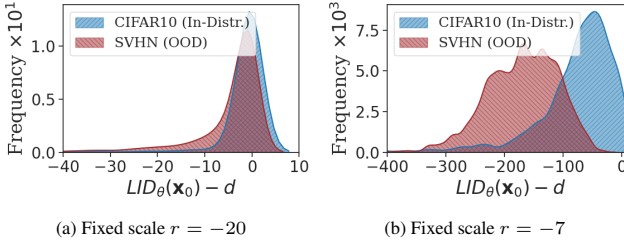

(a) Fixed scale $r = -20$      (b) Fixed scale $r = -7$

Figure 9: LID estimates, $\frac{\partial}{\partial r} \log \hat{\rho}_r(\mathbf{x}_0)$ for in-distribution (CIFAR10) and OOD data (SVHN): (a) at very negative value, LID estimates for both datasets overlap and are close to $d$; (b) at a larger value, OOD datapoints exhibit notably smaller LID estimates.

responds to the scale at which intrinsic dimension is assessed. Viewed at a small enough scale, a density will appear to imply a large intrinsic dimension, whereas, from a large enough scale, the entire local peak will appear to be a point. The goal here is to choose the best scale from which to differentiate OOD from in-distribution points. We note that this explanation is also why Tempczyk et al. (2022) use several noise scales.

One sensible way of setting $r$ is to calibrate it based on another model-free estimator of LID using the training data. In particular, we perform local principal component analysis (LPCA) which is a model-free method for LID estimation. LPCA is similar to our estimator in that it uses the concept of local linearizations. We use the `scikit-dimension` (Bac et al., 2021) implementation and use the algorithm introduced by Fukunaga & Olsen (1971) with `alphaFO` set to $0.001$ to estimate the average LID of our training data. Then $r$ is set so that $\text{LID}_\theta$ estimates of the training dataset match the LPCA average.

To increase efficiency, we select a random set of 80 data points from our training set as representative samples. We then employ a binary search to fine-tune $r$. During each iteration of the binary search, we compare the average $\text{LID}_\theta$ of our subsamples with the intrinsic dimension determined by LPCA. If the average $\text{LID}_\theta$ is lower, we increase $r$; otherwise, we decrease it. We initially set $r$'s binary search range between $-20$ and $20$, representing a wide range of plausible scales. Binary search is then executed in 20 steps to accurately ascertain a value of $r$. Table 2 represents three distinct scenarios to assess how to set $r$ optimally. In the first two rows, $r$ is held constant, while in the third, $r$ is dynamically adjusted based on the above approach. Although there is a minor performance drop in the FMNIST (vs) MNIST comparison, this is offset by a notable enhancement in the CIFAR10 (vs) SVHN case. This significant improvement justifies our preference for the adaptive method.

Table 2: Ablation study for the choice of $r$ evaluating on ROC-AUC (higher is better).

|  | FMNIST (vs) MNIST | MNIST (vs) FMNIST | CIFAR10 (vs) SVHN | SVHN (vs) CIFAR10 |
|---|---|---|---|---|
| $r = -20$ | **0.961** | 1.000 | 0.730 | **0.991** |
| $r = -10$ | 0.957 | 1.000 | 0.737 | 0.991 |
| Adaptive scaling | 0.953 | 1.000 | **0.926** | 0.986 |

Table 3: Essential hyperparameter settings for the normalizing flow models.

| Property | Model Configuration |
|---|---|
| Learning rate | $1 \times 10^{-3}$ |
| Gradient Clipping | Value based (max = 1.0) |
| Scheduler | `ExponentialLR` (with a factor of 0.99) |
| Optimizer | `AdamW` |
| Weight decay | $5 \times 10^{-5}$ |
| Batch size | 128 |
| Epochs | 400 |
| Transform blocks | Actnorm $\rightarrow$ $(1 \times 1)$ Convolution $\rightarrow$ Coupling |
| Number of multiscale levels | 7 levels |
| Coupling layer backbone | ResNet (channel size $= 64$, # blocks $= 2$, dropout $= 0.2$) |
| Masking scheme | Checkerboard |
| Latent Space | Standard isotropic Gaussian |
| Data pre-processing | Dequantization & Logit scaling |
| Data shape | $28 \times 28 \times 1$ for grayscale and $32 \times 32 \times 3$ for RGB |

## D.2 HYPERPARAMETER SETTING FOR NORMALIZING FLOWS

We trained both Glow (Kingma & Dhariwal, 2018) and RQ-NSFs (Durkan et al., 2019) on our datasets, with the hyperparameters detailed in Table 3. Specifically, while Glow utilized an affine coupling layer, we adopted RQ-NSF's piecewise rational quadratic coupling with two bins and linear tails capped at 1. In Figure 11 and Figure 12, we highlight the Glow architecture's failure cases. The artifacts, particularly in CelebA, Tiny ImageNet, and Omniglot samples, stem from the affine coupling layers' unfavourable numerical properties. In contrast, the RQ-NSF architectures showed no such issues, leading us to adopt them for subsequent experiments.

In the context of OOD detection, expressive architectures sometimes face issues of numerical non-invertibility and exploding inverses, particularly with OOD samples. Behrmann et al. (2021) argue that while expressive NFs adeptly fit data manifolds, their mapping from a full-dimensional space to a lower-dimensional one can cause non-invertibility, especially in OOD datapoints. They specifically identified non-invertibility examples in Glow models on OOD data. Contrarily, the RQ-NSFs we trained according to the hyperparameter setup in Table 3 demonstrated full reconstruction on OOD data, as depicted in Figure 10. This is another reason why we chose RQ-NSFs.

We used an NVIDIA Tesla V100 SXM2 with 7 hours of GPU time to train each of the models.

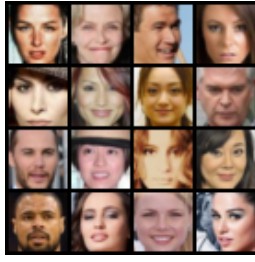 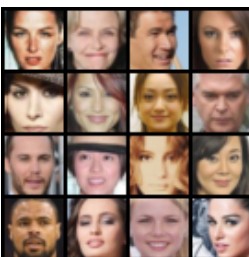 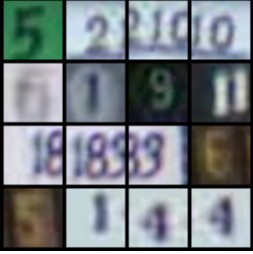 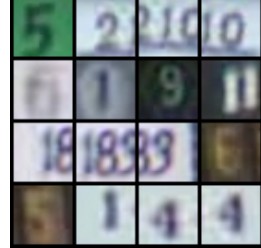

(a) Original samples from the test split.  (b) Reconstructed samples from the test split.  (c) Original samples from the OOD dataset.  (d) Reconstructed samples from the OOD dataset.

Figure 10: Numerical invertibility: (a-b) A random batch of samples and their reconstructions from the test split of an RQ-NSF model trained on CelebA. (c-d) A random batch of samples and their reconstructions from the OOD dataset, SVHN.

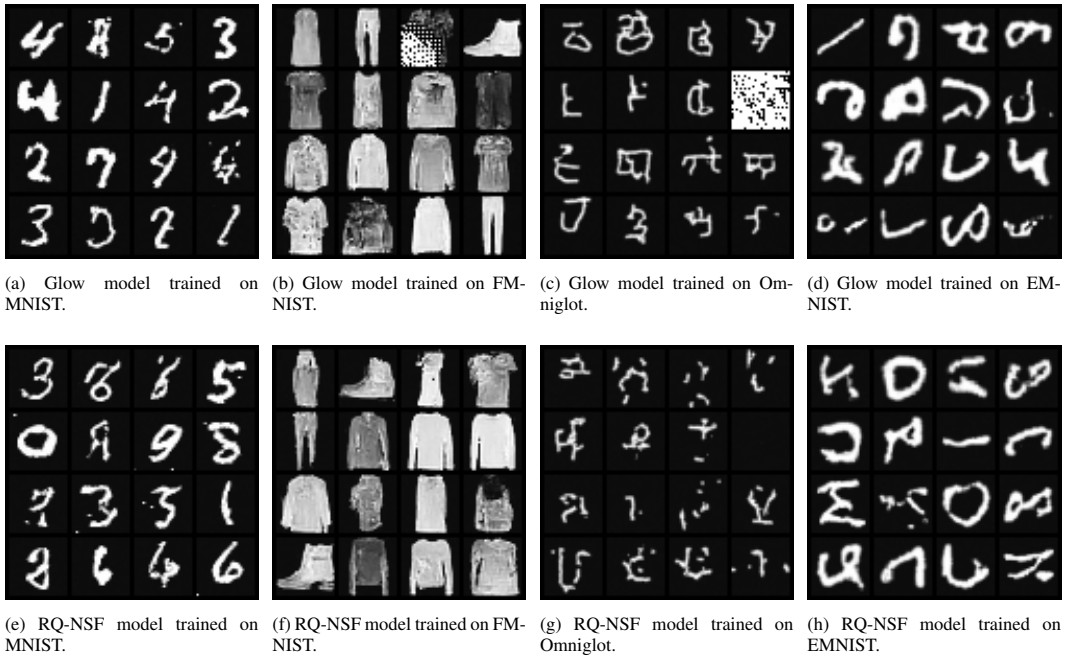

(a) Glow model trained on MNIST.

(b) Glow model trained on FMNIST.

(c) Glow model trained on Omniglot.

(d) Glow model trained on EMNIST.

(e) RQ-NSF model trained on MNIST.

(f) RQ-NSF model trained on FMNIST.

(g) RQ-NSF model trained on Omniglot.

(h) RQ-NSF model trained on EMNIST.

Figure 11: Samples generated from models trained on the grayscale collection: due to numerical properties of affine coupling layers, Glow models tend to produce artifacts in their generated data.

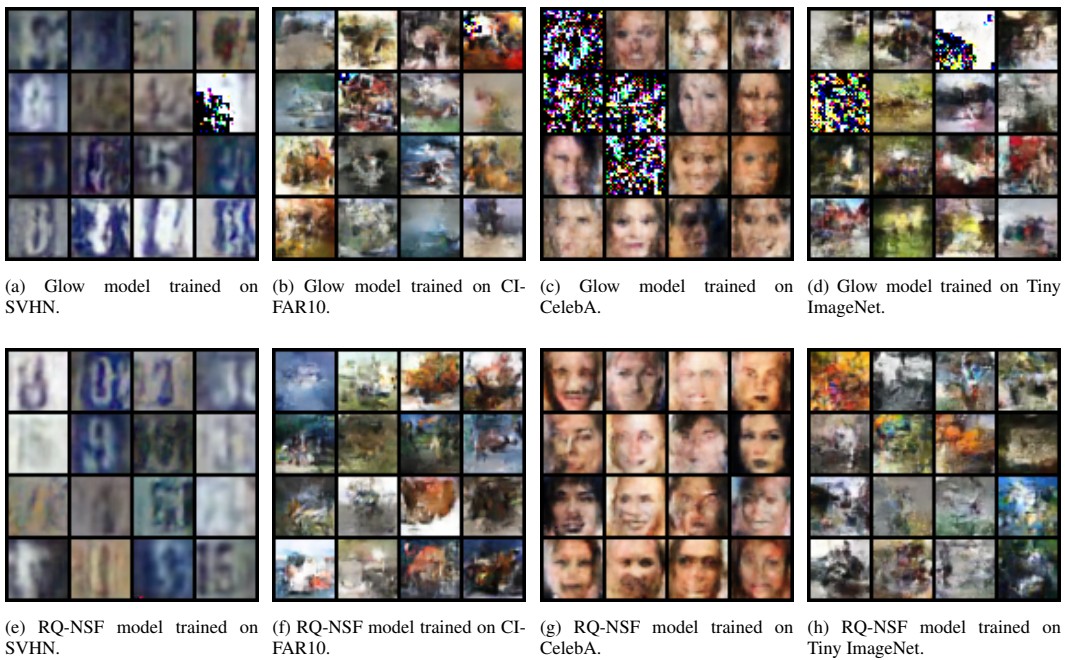

(a) Glow model trained on SVHN.

(b) Glow model trained on CIFAR10.

(c) Glow model trained on CelebA.

(d) Glow model trained on Tiny ImageNet.

(e) RQ-NSF model trained on SVHN.

(f) RQ-NSF model trained on CIFAR10.

(g) RQ-NSF model trained on CelebA.

(h) RQ-NSF model trained on Tiny ImageNet.

Figure 12: Samples generated from models trained on the RGB collection: the artifacts in Glow models are apparent.

### D.3 INTRINSIC DIMENSION VS LIKELIHOOD SCATTERPLOTS

In this section, we introduce scatterplots analogous to those in Figure 4 that illustrate results for various tasks. As emphasized previously, we consistently observe that LID estimates for pathological OOD data points are generally lower than the in-distribution data. This confirms our hypothesis suggesting that these points are situated on a lower dimensional submanifold that is assigned minimal probability mass. This behaviour is evident across all figures in this section that exhibit pathological likelihoods. Examples include MNIST-gen (vs) FMNIST and Omniglot-gen (vs) EMNIST on the right, and all the figures on the left. Conversely, in non-pathological scenarios (shown on the right), OOD data points are generally assigned lower likelihoods. Our dual thresholding approach effectively identifies both scenarios by simultaneously setting thresholds on both $\log p_\theta(\mathbf{x}_0)$ and $\mathrm{LID}_\theta(\mathbf{x}_0)$. Results are shown in Figure 13, Figure 14, Figure 15, and Figure 16.

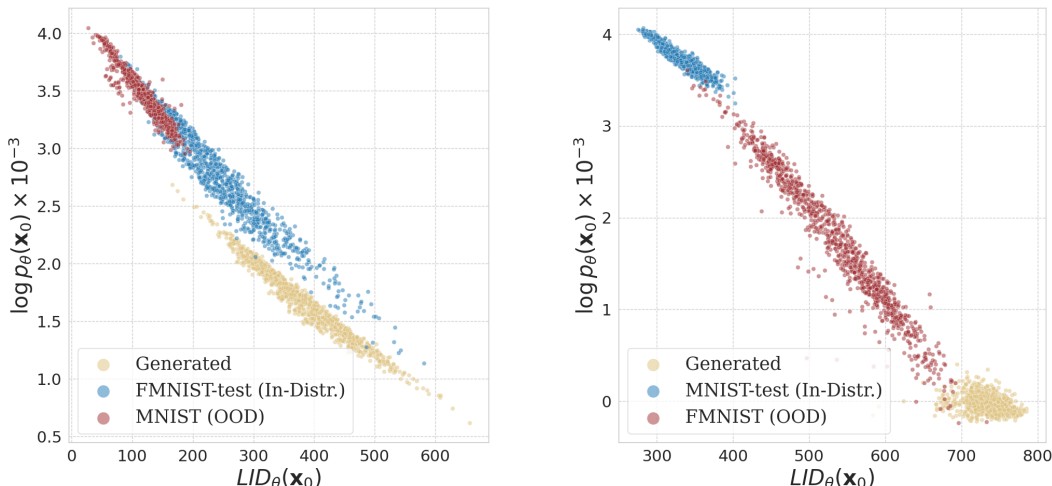

Figure 13: Visualization of LID estimates using adaptive scaling and likelihoods per datum where the $x$-axis represents $\mathrm{LID}_\theta(\mathbf{x}_0)$ and the $y$-axis represents $\log p_\theta(\mathbf{x}_0)$. **Left:** Scatterplots for the pathological OOD detection tasks "FMNIST (vs) MNIST" and "FMNIST-gen (vs) MNIST". **Right:** Scatterplots for the non-pathological OOD detection tasks "MNIST (vs) FMNIST" and "MNIST-gen (vs) FMNIST".

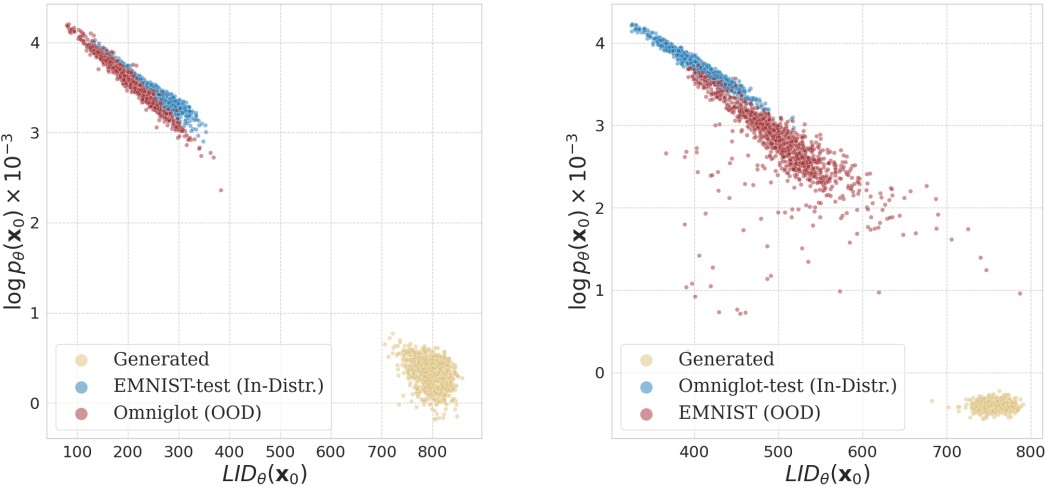

Figure 14: Visualization of LID estimates using adaptive scaling and likelihoods per datum where the $x$-axis represents $\mathrm{LID}_\theta(\mathbf{x}_0)$ and the $y$-axis represents $\log p_\theta(\mathbf{x}_0)$. **Left:** Scatterplots for the pathological OOD detection tasks "EMNIST (vs) Omniglot" and "EMNIST-gen (vs) Omniglot". **Right:** Scatterplots for the non-pathological OOD detection tasks "Omniglot (vs) EMNIST" and "Omniglot-gen (vs) EMNIST".

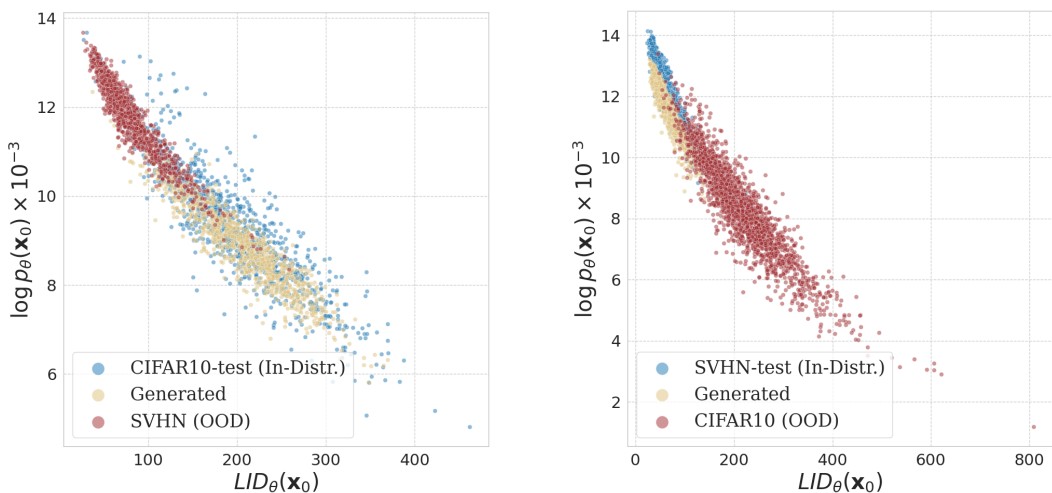

Figure 15: Visualization of LID estimates using adaptive scaling and likelihoods per datum where the $x$-axis represents $\text{LID}_\theta(\mathbf{x}_0)$ and the $y$-axis represents $\log p_\theta(\mathbf{x}_0)$. **Left:** Scatterplots for the pathological OOD detection tasks "CIFAR10 (vs) SVHN" and "CIFAR10-gen (vs) SVHN". **Right:** Scatterplots for the non-pathological OOD detection tasks "SVHN (vs) CIFAR10" and "SVHN-gen (vs) CIFAR10"

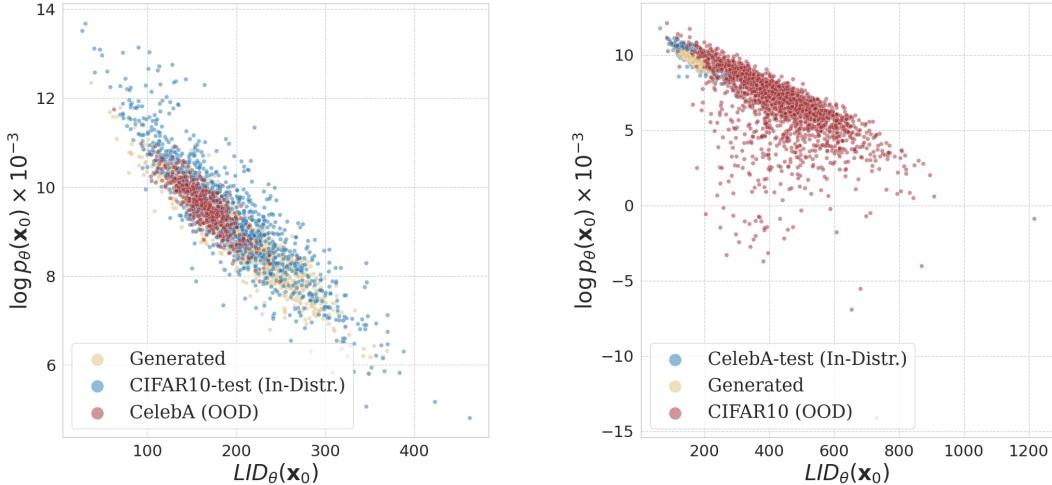

Figure 16: Visualization of LID estimates using adaptive scaling and likelihoods per datum where the $x$-axis represents $\text{LID}_\theta(\mathbf{x}_0)$ and the $y$-axis represents $\log p_\theta(\mathbf{x}_0)$. **Left:** Scatterplots for the pathological OOD detection tasks "CIFAR10 (vs) CelebA" and "CIFAR10-gen (vs) CelebA". **Right:** Scatterplots for the non-pathological OOD detection tasks "CelebA (vs) CIFAR10" and "CelebA-gen (vs) CIFAR10".

Table 4: ROC-AUC (higher is better) at A-gen (vs) B tasks. **Notation**: * tasks where likelihoods alone do not exhibit pathological behaviour, ‡ methods that employ external information or auxiliary models. For each task, we bold the best performing model.

| Trained on | MNIST * | | FMNIST | | CIFAR10 | | SVHN * | |
|---|---|---|---|---|---|---|---|---|
| OOD Dataset | FMNIST | Omniglot | MNIST | Omniglot | SVHN | CelebA | CIFAR10 | CelebA |
| Naïve Likelihood $\log p_\theta(\mathbf{x}_0)$ | 0.004 | 0.000 | 0.040 | 0.000 | 0.025 | 0.240 | 0.971 | **0.985** |
| $\|\frac{\partial}{\partial \mathbf{x}} \log p_\theta(\mathbf{x}_0)\|_2$ | 0.997 | 0.997 | 0.993 | 0.993 | 0.712 | 0.379 | 0.195 | 0.077 |
| Complexity Correction‡ | 0.026 | 0.000 | 0.044 | 0.001 | 0.678 | 0.243 | 0.714 | 0.451 |
| Likelihood Ratios‡ | 0.998 | **1.000** | **1.000** | **1.000** | 0.299 | 0.396 | 0.302 | 0.099 |
| Dual Threshold (Ours) | **0.999** | **1.000** | **1.000** | 0.996 | **0.951** | **0.733** | **0.970** | **0.985** |

## D.4 Additional AUC Results

Table 4 compares methods by seeing how well they can differentiate between generated samples from $p_\theta$ and OOD data. Overall, our dual threshold method performs consistently well across tasks.

Table 5 compares naïve likelihoods against our dual threshold method across all tasks. We can see an extremely consistent improvement, highlighting the relevance of dual thresholding.

## D.5 Extra Ablations

Throughout our paper, we have argued in favour of our dual threshold method, which combines likelihoods and LID estimates. To highlight that our strong performance is not just based on dual thresholding itself, we carry out an ablation where we use dual thresholding, but on likelihood and gradient norm $\|\frac{\partial}{\partial \mathbf{x}} \log p_\theta(\mathbf{x}_0)\|_2$ pairs. Table 6 shows the results, highlighting that LID estimates are much more useful. The table also shows that using single thresholds with LID estimates is also not enough to reliably detect OOD points.

Table 5: ROC-AUC (higher is better) using likelihoods only, compared to dual thresholds. The top part of the table contains greyscale tasks, and the bottom contains RGB tasks. Values are bolded when there is at least a 20% improvement from dual thresholding over likelihoods alone.

| OOD Task Type | $A$-gen (vs) $B$ [§] | | $A$ (vs) $B$ [†] | |
|---|---|---|---|---|
| Dataset Pair $A$ (and) $B$ | (AUC-ROC) Naïve Likelihood | (AUC-ROC) Dual Threshold | (AUC-ROC) Naïve Likelihood | (AUC-ROC) Dual Threshold |
| EMNIST (and) MNIST | **0.000** | **1.000** | **0.533** | **0.797** |
| EMNIST (and) Omniglot | **0.000** | **1.000** | **0.397** | **0.814** |
| EMNIST (and) FMNIST | **0.039** | **1.000** | 0.998 | 0.998 |
| Omniglot (and) MNIST | **0.011** | **1.000** | 1.000 | 1.000 |
| Omniglot (and) EMNIST | **0.000** | **1.000** | 0.983 | 0.983 |
| Omniglot (and) FMNIST | **0.138** | **1.000** | 1.000 | 1.000 |
| FMNIST (and) MNIST | **0.000** | **1.000** | **0.070** | **0.953** |
| FMNIST (and) EMNIST | **0.001** | **0.960** | **0.391** | **0.605** |
| FMNIST (and) Omniglot | **0.000** | **0.996** | **0.086** | **0.862** |
| MNIST (and) EMNIST | **0.000** | **1.000** | 0.985 | 0.985 |
| MNIST (and) Omniglot | **0.000** | **1.000** | **0.787** | **0.869** |
| MNIST (and) FMNIST | **0.005** | **0.999** | 1.000 | 1.000 |
| CelebA (and) Tiny | 0.933 | 0.965 | 0.905 | 0.936 |
| CelebA (and) SVHN | **0.154** | **0.930** | **0.151** | **0.949** |
| CelebA (and) CIFAR100 | 0.946 | 0.967 | 0.919 | 0.941 |
| CelebA (and) CIFAR10 | 0.944 | 0.965 | 0.915 | 0.935 |
| Tiny (and) CelebA | 0.640 | 0.646 | 0.812 | 0.812 |
| Tiny (and) SVHN | **0.036** | **0.951** | **0.164** | **0.907** |
| Tiny (and) CIFAR100 | **0.686** | **0.776** | 0.796 | 0.822 |
| Tiny (and) CIFAR10 | **0.691** | **0.767** | 0.802 | 0.825 |
| SVHN (and) CelebA | 0.984 | 0.984 | 0.995 | 0.995 |
| SVHN (and) Tiny | 0.971 | 0.971 | 0.987 | 0.987 |
| SVHN (and) CIFAR100 | 0.970 | 0.970 | 0.987 | 0.987 |
| SVHN (and) CIFAR10 | 0.970 | 0.970 | 0.986 | 0.986 |
| CIFAR100 (and) CelebA | **0.225** | **0.646** | **0.378** | **0.635** |
| CIFAR100 (and) Tiny | **0.392** | **0.453** | 0.485 | 0.493 |
| CIFAR100 (and) SVHN | **0.017** | **0.941** | **0.076** | **0.930** |
| CIFAR100 (and) CIFAR10 | **0.402** | **0.502** | 0.492 | 0.502 |
| CIFAR10 (and) CelebA | **0.258** | **0.733** | **0.413** | **0.653** |
| CIFAR10 (and) Tiny | **0.445** | **0.555** | 0.543 | 0.548 |
| CIFAR10 (and) SVHN | **0.017** | **0.951** | **0.069** | **0.926** |
| CIFAR10 (and) CIFAR100 | **0.424** | **0.610** | 0.518 | 0.561 |

Table 6: Ablation study for the dual thresholding method on ROC-AUC (higher is better).

| Method | FMNIST (vs) MNIST | MNIST (vs) FMNIST | CIFAR10 (vs) SVHN | SVHN (vs) CIFAR10 |
|---|---|---|---|---|
| $\text{LID}_\theta(\mathbf{x}_0)$ | 0.947 | 0.002 | 0.949 | 0.009 |
| $\|\frac{\partial}{\partial \mathbf{x}} \log p_\theta(\mathbf{x}_0)\|_2 + \log p_\theta(\mathbf{x})$ | 0.509 | 0.983 | 0.716 | 0.962 |
| $\text{LID}_\theta(\mathbf{x}_0) + \log p_\theta(\mathbf{x})$ (Ours) | **0.953** | **1.000** | **0.926** | **0.986** |

# E    CRITICAL ANALYSIS OF OOD BASELINES

As we outlined in Section 5, when benchmarking against the complexity correction and likelihood ratio methods, we observed notable underperformance in non-pathological directions. Both methods aim to correct inflated likelihoods encountered in pathological OOD scenarios by assigning a score to each datapoint which is obtained by adding a complexity term to the likelihood (Serrà et al., 2020), or subtracting a reference likelihood obtained from a model trained on augmented data (Ren et al., 2019). This score then becomes the foundation for their OOD detection through single thresholding. However, as we will demonstrate in this section, these techniques often necessitate an artificial hyperparameter setup to combine these metrics together, making it less than ideal.

Formally, both of these studies aim to find a score $S(\mathbf{x}_0)$ to correct the inflated likelihood term $\log p_\theta(\mathbf{x}_0)$, by adding a metric to $m(\mathbf{x}_0)$ as follows:

$$S(\mathbf{x}_0) := \log p_\theta(\mathbf{x}_0) + \lambda \cdot m(\mathbf{x}_0), \tag{12}$$

In Serrà et al. (2020), $\lambda = 1$ and $m(\mathbf{x}_0)$ is the bit count derived by compressing $\mathbf{x}_0$ using three distinct image compression algorithms and selecting the least bit count from the trio (an ensemble approach as they describe). The algorithms include standard `cv2`, PNG, JPEG2000, and FLIF (Sneyers & Wuille, 2016). Moreover, we did not find any official implementation for the complexity correction method; however, since their algorithm was fairly straightforward, we re-implemented it according to their paper and it is readily reproducible in our experiments. On the other hand, Ren et al. (2019) propose training a reference likelihood model with the same architecture as the original model; however, on perturbed data. We employ another RQ-NSF, samples of which are depicted in the bottom row of Figure 17. Ren et al. (2019) claim that their reference model only learns background statistics that are unimportant to the semantics we care for in OOD detection; hence, subtracting the reference likelihood $m(\mathbf{x}_0)$ can effectively correct for these confounding statistics that potentially inflate our original likelihoods. That being said, they employ a complicated hyperparameter tuning process on $\lambda$ to ensure best model performance.

As illustrated in Figure 18, we sweep values of $\lambda$ and compare our method against all these models. While certain $\lambda$ values enhance OOD detection in pathological scenarios, they falter in non-pathological contexts. In contrast, our dual thresholding remains robust irrespective of the scenario's nature. This observation underscores a significant gap in the OOD detection literature. While several methods address the OOD detection pathologies, many are overly specialized, performing well predominantly in the pathological direction. We note that the results we report in Table 1 correspond to the best values of $\lambda$.

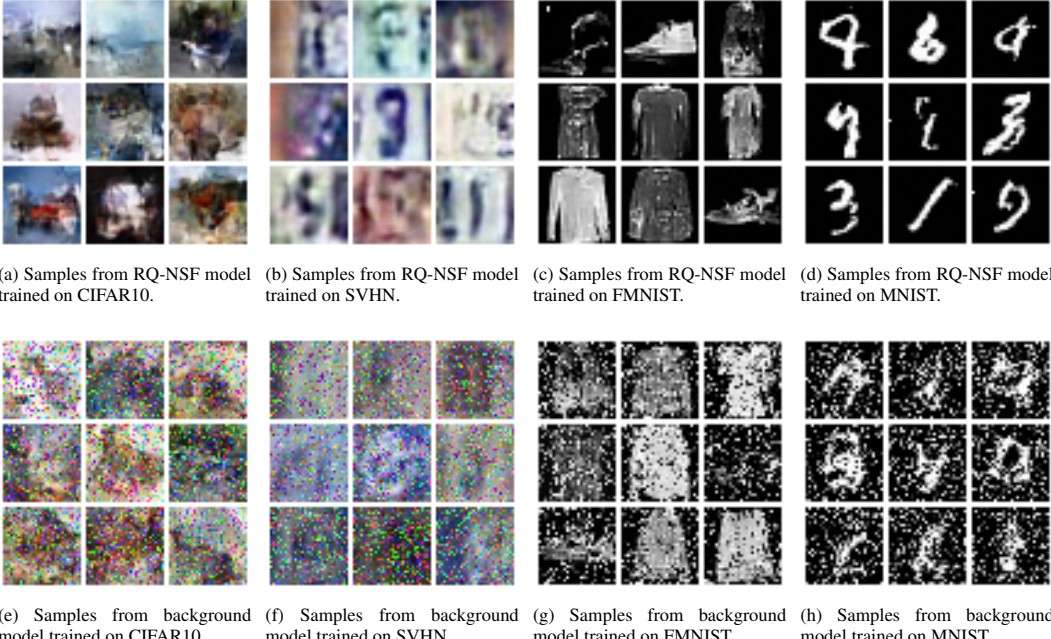

(a) Samples from RQ-NSF model trained on CIFAR10.

(b) Samples from RQ-NSF model trained on SVHN.

(c) Samples from RQ-NSF model trained on FMNIST.

(d) Samples from RQ-NSF model trained on MNIST.

(e) Samples from background model trained on CIFAR10.

(f) Samples from background model trained on SVHN.

(g) Samples from background model trained on FMNIST.

(h) Samples from background model trained on MNIST.

Figure 17: Samples generated from normal and background models that are trained using the RQ-NSF hyperparameters provided in Table 3. The background models are trained on perturbed data, using the scheme presented by Ren et al. (2019).

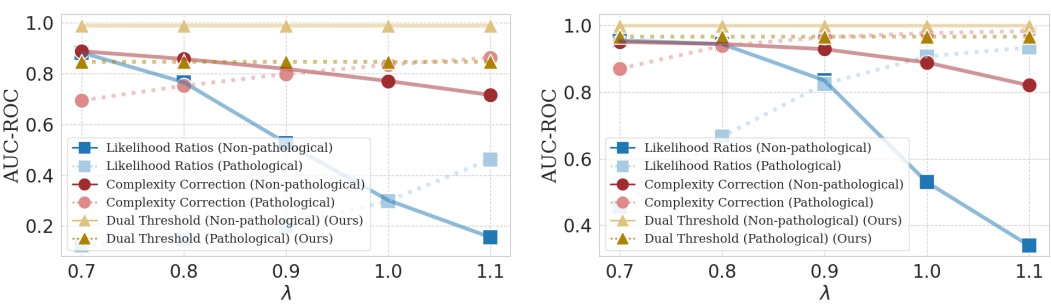

(a) Performance comparison of different methods on two pathological and non-pathological OOD detection tasks obtained from the FMNIST and MNIST pair.

(b) Performance comparison of different methods on two pathological and non-pathological OOD detection tasks obtained from the CIFAR10 and SVHN pair.

Figure 18: Comparing our dual thresholding approach of combining metrics to all the different single score thresholding baselines by sweeping over different values of $\lambda$ in Equation 12. The tasks that are considered are either: (i) pathological such as "FMNIST (vs) MNIST" (left) or "CIFAR10 (vs) SVHN" (right); or (ii) non-pathological such as "MNIST (vs) FMNIST" (left) or "SVHN (vs) CIFAR10" (right).

