# OpenReview forum: "Explaining the Out-of-Distribution Detection Paradox through Likelihood Peaks"
_ICLR.cc/2024/Conference — Submitted to ICLR 2024_

### Official Review · Reviewer_i5V7 · 2023-10-30

**Soundness:** 2 fair
**Presentation:** 2 fair
**Contribution:** 3 good
**Rating:** 6
**Confidence:** 4

**Summary:**

The paper proposes a new method for out-of-distribution detection using a normalizing flow. The method is motivated by the Local Intrinsic Dimension estimation using approximate Likelihood (LIDL) and at a high level rejects a datapoint as OOD if either its density or local intrinsic dimension is low. The paper presents an approximation to LIDL that allows it to be applicable given just a single normalizing flow. The authors choose hyperparameters for their approach by considering SVHN vs. CIFAR and the reverse as validation tasks and show good OOD detection performance across eight different image dataset pairs.

**Strengths:**

1. Originality: The paper is the first to my knowledge to use an estimate of local intrinsic dimension to detection OOD datapoints. The authors come up with an approximation to a recent estimation technique for LID that allows it to be utilized with a single normalizing flow.
2. Quality: The authors present great visualizations and and good supplemental analyses to the main results.
3. Clarity: The method and experiments are clearly written.
4. Significance: The authors address an interesting problem and propose a new angle.

**Weaknesses:**

Soundness:
1. The approximation of $\log \rho_r(\mathbf{x})$ comes about by using a first-order Taylor approximation of the invertible function $f_\theta$ defining the flow. It's not clear to me how this upstream approximation affects the downstream approximation of the actual quantity of interest, $\rho_r(\mathbf{x})$. In addition, the fact that $\hat{\rho}_r(\mathbf{x})$ is shown to be a poor approximation of $\rho_r(\mathbf{x})$ in "Identifying Sharp Likelihood Peaks" (in that the order for ID and OOD data is the opposite of what is intended) makes it a bit tenuous why it's ok for the $\hat{\rho}_r(\mathbf{x})$ to be used to approximate $\rho_r(\mathbf{x})$ in $\frac{\partial}{\partial r}\log \rho_r(\mathbf{x})$.
2. The proposed method is sensitive to choice of $r$, yielding almost no separation between datasets under some values of $r$. The authors instead set hyperparameters using two of the dataset pairs, CIFAR-10 vs. SVHN and the reverse, yet these two benchmarks are also two of the eight benchmarks presented in the main results, and at least one of the datasets is in four of the eight benchmarks.

Presentation:
1. It would benefit the authors to be a bit more careful in their discussion about manifolds and sub-manifolds. Namely, it is a bit strange to hypothesize that OOD data lies on a lower-dimensional submanifold than ID data and call these "lower probability mass" regions, since a submanifold constitutes a measure-zero set with respect to the Lebesgue measure of the ambient dimension of the data. At one point the authors use the phrase "concentrate around a lower-dimensional submanifold" which might be better, but it would be helpful to formalize what the authors mean so that the hypothesis makes mathematical sense.
2. The first contribution claimed by the paper (to explain how DGMs can assign higher likelihood to OOD datapoints yet only generate in-distribution samples) has already been offered by previous work. See section 5.1 in [1], which offers the same explanation as the one given in this current submission, i.e. "OOD datapoints can be assigned higher likelihood while not being generated if they belong to regions of low probability mass." Moreover, in the related work, the authors repeat arguments made in [1] as if they were their own, e.g. "...we challenge this assumption..."
3. In Sec 4.1, the authors start by considering a uniform distribution over the volume of a given R-radius ball but then switch mid-paragraph to considering a uniform distribution on just the surface of said ball. It would help to explicitly mention the phenomenon that in high dimensions nearly all the volume is at the surface to justify the jump.

[1] Zhang et al. 2021. Understanding Failures in Out-of-Distribution Detection with Deep Generative Models. ICML 2021.

**Questions:**

1. Do the authors have any analysis to suggest that $\hat{\rho}_r(\mathbf{x})$ should be a good approximation of $\rho_r(\mathbf{x})$?
2. Can the authors provide evidence that this method works better than, say, an arbitrary statistic of the generative model that is validated to perform well on both SVHN vs. CIFAR-10 and the reverse? How about that the proposed method is robust to the validation strategy (e.g. alternative dataset pairs or different strategy altogether)?

I think the paper has the potential to yield an interesting contribution if at least one of the above questions is addressed. In such a scenario, provided the above weaknesses are also addressed, I would be happy to raise my score.

---

> ### Author Response · Authors · 2023-11-17
> **Response to Reviewer i5V7**
>
> Thank you for the insightful review and praise for our ideas and presentation. We address your concerns point-by-point below:
>
> Soundness
> 1. You expressed some concerns about the validity of our Taylor approximation of $\log \rho_r(\mathbf{x})$. Part of this was based on how $\hat{\rho}_r$ ranks in-distribution and OOD data in Figure 3 in the manuscript. In fact, Figure 3 shows them behaving precisely as would be expected. For very negative $r$, the OOD data has higher mass because it is assigned higher likelihood under the model. For values closer to zero (here above $r \approx -7$), the Taylor approximation loses its accuracy since the mass is estimated over too large of a region. The slopes of the curves are also appropriate, as the more negative slope for OOD data indicates it has lower LID. To further alleviate these concerns, we are conducting experiments to verify the suitability of our LID estimator on benchmark datasets, and we will include them before the end of the discussion period.
> 2. You shared a concern about how we set $r$ with reviewer **urvk**; please find our response to this in the general rebuttal.
>
> Presentation
> 1. You asked us to be more precise in the work about manifolds, submanifolds, and probability mass. Thank you for the suggestion; we will update the language around manifolds in the manuscript accordingly. To summarize,
>    - The in-distribution data sits on a submanifold (of zero Lebesgue measure).
>    - The likelihood-based DGM, which represents a full-dimensional density, concentrates mass around the (complex) in-distribution submanifold in such a way as to encode the submanifold’s dimensionality [A].
>    - Due to its inductive biases, the DGM density *also* inadvertently concentrates a negligible amount of mass around out-of-distribution submanifolds when these are “simpler” (i.e. of lower intrinsic dimension).
>
>     Since there is a distinction between the probability mass of the DGM and the underlying (measure zero) submanifolds being modelled, there is no underlying error in the way we use these concepts, though we certainly take your point that they can be made clearer.
> 2. We thank you for pointing this out. You are correct that the notion that the OOD paradox must be due to lower probability mass was not originally proposed by us, although we do highlight that the paper you point out does not propose a way to empirically verify this. We will reword the manuscript to make this clear. Our main contributions are linking this explanation to intrinsic dimension, and exploiting this newfound connection to build a highly-performant OOD detection algorithm.
> 3. You suggested a rewording to make the jump from a uniform distribution on a high-dimensional ball to a Gaussian a bit clearer. Thank you for this; we will clarify this paragraph in the manuscript.
>
> You also had two additional questions:
>
> 1. Thank you for bringing this point up. We are currently conducting a set of experiments where the ground truth values of $\rho_r$ and $\frac{\partial}{\partial r} \rho_r$ are either known or can be accurately computed. We will include these results in the manuscript before the end of the discussion period.
> 2. You also asked us about whether choosing hyperparameters for $r$ based on SVHN-CIFAR-10 does better than choosing another arbitrary statistic from a generative model, and whether our method is robust to this choice. In the general rebuttal we propose a new intuitive way to select $r$ without hyperparameter tuning; we hope this resolves your concerns on this front. Otherwise, could you please clarify what kind of arbitrary statistic you have in mind? We do point out that the “complexity correction” and “likelihood ratios” baselines we compare against could be loosely interpreted as leveraging arbitrary statistics, namely the compression length of images and the likelihood of another model, respectively.
>
> [A] Tempczyk et al., “LIDL: Local Intrinsic Dimension Estimation Using Approximate Likelihood”, ICML 2022.

---

> > ### Author Response · Authors · 2023-11-19
> > **Validating the accuracy of $\hat{\rho}_r$**
> >
> > In response to your concerns about our approximation $\log \hat{\rho}_r$, we have designed an experiment using synthetic datasets to check its accuracy. Our experiment shows that $\log \hat{\rho}_r$ accurately represents $\log\rho_r$, the logarithm of a Gaussian convolved with a model’s density.
> >
> > First, we generated an 800-dimensional dataset from a mixture of four Gaussians with different means and covariances. After training an RQ-NSF on 70,000 samples of this distribution, we obtain a $p_\theta$ that fits this mixture of Gaussians well. The modes of the distribution are set far apart and their componentwise covariance matrices are set so that there is minimal overlap between them; this setting allows us to derive an analytical expression to accurately approximate the density around each mode as follows. If $c_i$ is the probability of a sample belonging to the $i$th component with $\sum_i c_i=1$, and $\mathbf{x}_i$ and $\mathbf{A}_i$ represent the mean and covariance matrix of the $i$th component, respectively,  then the density around each mode can be calculated as if the other components do not exist:
> >
> > \begin{equation}
> > \log p_\theta(\mathbf{x}_i) \approx \log c_i \cdot \mathcal{N}(\mathbf{x}_i;  \mathbf{x}_i, \mathbf{A}_i) \Rightarrow \log \rho_r(\mathbf{x}_i)  \approx \log \frac{1}{\sqrt{(2\pi)^d \det(\mathbf{A}_i + e^{2r} \mathbf{I})}}
> > \end{equation}
> >
> > Therefore, comparing our estimator inspired by the Taylor expansion with the value obtained above provides us with a way to validate the quality of our estimator for $\log \rho_r$. Figure 2 (a-d) of our newly uploaded supplementary material shows the estimated $\log \hat{\rho}_r$ for every mode vis-à-vis the accurate value obtained analytically from the above equation as $r$ increases. This comparison confirms that these estimators are accurate approximations.

---

> > > ### Author Response · Authors · 2023-11-21
> > > **Theoretical results on the quality of the estimator $\hat{\rho}_r$:**
> > >
> > > In addition to our empirical results described above and featured in the supplementary material, we have also included an absolute error bound for $\hat{\rho}_r$. Please check the section titled “On the Quality of the Linear Approximation” in the updated supplementary material for formal discussion and proofs.
> > >
> > > Recall that in subsection 4.1 of the paper, we use Taylor expansions to establish an approximation of for the density: $\hat{p}(\mathbf{x}) = N(\mathbf{x}; \mathbf{x}_0 - \mathbf{J}_0 \mathbf{z}_0, \mathbf{J}_0 \mathbf{J}_0^T + e^{2r} \mathbf{I}_d)$.
> > > In our newly added discussion, we claim that even if the Taylor approximation is only accurate in a small radius around $\mathbf{x}_0$, the downstream estimate $\hat{\rho}_r(\mathbf{x}_0)$ still provides a good approximation for $\rho_r(\mathbf{x}_0).$ Formally, assume that for a given small radius $R$, we have that
> > >
> > > $$|\hat{p}(\mathbf{x}_0) - p(\mathbf{x}_0)| \le e(R)$$
> > >
> > > where $e(R)$ is the local (around $\mathbf{x}_0$) approximation error in the density incurred by the Taylor approximation. Now, the absolute error between the estimator $\hat{\rho}_r(\mathbf{x}_0)$ and the actual value $\rho_r(\mathbf{x}_0)$ can be bounded as follows:
> > >
> > > $$|\rho_r(\mathbf{x}_0) - \hat{\rho}_r(\mathbf{x}_0)| = e(R) + O\left( \left[ \frac{R}{d e^r} e^{1 - \frac{R}{de^r}}\right]^{d/2} \right).$$
> > >
> > > Therefore, as $r$ decreases, the error margin for $\hat{\rho_r}$ approaches that of the original approximation $\hat{p}$ exponentially fast.

---

> > ### Comment · Reviewer_i5V7 · 2023-11-22
> > **Thanks for the reply**
> >
> > Thank you for the response. For the points related to soundness, I appreciate that the authors have added an error bound for the estimator $\hat{\rho}$ and updated the hyperparameter selection of $r$. One follow-up question to the latter: how is $k$ selected for LPCA? A quick note on the former: it looks like there's a small typo in the Lemma 1.1 proof, that the domain of integration hasn't been updated with the change of variables.
> >
> > On presentation, I appreciate the author's responses and look forward to the updated main paper; should I expect this by the end of the discussion period?
> >
> > On the questions, thank you for addressing the first with the synthetic experiment and error bound. The second is related to the concerns I had about the original selection of $r$ using the CIFAR-10 vs. SVHN and SVHN vs. CIFAR-10 OOD detection tasks, which could lead to the scenario of fitting the method to the detection tasks themselves. With the updated selection of $r$, I am less concerned about this issue, and with evidence that the estimator $\hat{\rho}$ can be a good approximation of $\rho$, I view the updated experimental results more favorably as support for the paper's main hypothesis. As such, I have raised my score.

---

> > > ### Author Response · Authors · 2023-11-22
> > > **Thank you!**
> > >
> > > Thank you for raising your score! We are working on the revised manuscript now and will have it in by the end of the discussion period.
> > >
> > > - As for LPCA, we used the default hyperparameters in the scikit-dim implementation [A], except for alphaFO=0.001. (The default of 0.05 gives extremely low dimensionality estimates across the board, so we used a smaller value. It was not chosen with respect to any particular dataset or metric.)
> > >
> > > - Upon double-checking, we believe the change in the domain of integration is correct. Since $\mathbf{v}$ is a rescaling of $\mathbf{u}$ by $\sigma^{-1}$, the ball’s radius is also rescaled by $\sigma^{-1}$ (though admittedly this is difficult to catch because it is shown in the ball’s subscript).
> > >
> > >
> > > [A] https://scikit-dimension.readthedocs.io/en/stable/skdim.id.lPCA.html

---

> > > > ### Comment · Reviewer_i5V7 · 2023-11-23
> > > > **Re: Thank you!**
> > > >
> > > > Thanks for the reply. Re: point 2, you're right; sorry, I mistakenly read the domain of integration of the two lines as the same expression and did indeed miss the dropped $\sigma$.

---

### Official Review · Reviewer_c1to · 2023-10-31

**Soundness:** 2 fair
**Presentation:** 3 good
**Contribution:** 2 fair
**Rating:** 5
**Confidence:** 1

**Summary:**

This paper tried to explain the paradox "OOD samples are never generated by these DGMs despite having
high likelihoods" and propose using the probability mass that the model assigns to a small neighborhood of a data sample as the OOD detection method.

**Strengths:**

This paper analyzes the availability and efficiency of using LID estimators for OOD detection and proposes to use density-based methods instead of the previous data-based ones. Specifically, it is inspired by a previous observation of adding Gaussian noise to corrupt the density, which is easy to understand but low-efficiency in computing as multiple NFs are needed, the authors propose a much more efficient method that only needs one pre-trained NF.

**Weaknesses:**

**W1:** The claim **"This observation becomes even more puzzling in light of the fact that said DGMs generate
only in-distribution samples..."**, which is also the major motivation of this work, could be wrong. Though existing research always takes a different dataset like SVHN as an OOD dataset to quantitatively evaluate the model's OOD detection performance when trained on IID dataset like CIFAR-10, we should not only focus on these real-world datasets. For example, when drawing multiple generations out of a deep generative model, some of the generation samples could be very different and low-quality compared to the IID training set, where these samples should also be seen as OOD samples. When considering these OOD samples, the claim could be wrong. Overall, if the generative model cannot exactly model $p(x_{IID})$, there is no guarantee that a deep generative model would never generate OOD samples, otherwise there would be no such "puzzling behavior".

**W2:** There are already some theoretical explanations for the flow-based generative model's over-confidence in OOD samples, such as the location and variances of the data and the model curvature [1]. Thus, what is the difference between their interpretation and the explanation in this paper?

[1] Do Deep Generative Models Know What They Don't Know? ICLR 2019.

**W3:** The proposed method highly depends on a strong assumption that the probability mass around an OOD sample is negligible. However, the authors seem not to provide sufficient support for this assumption, which may weaken the convincing of the proposed methods. And the experiments can also not support this assumption well, as the tested datasets are limited. Actually, not all OOD datasets would be assigned higher likelihoods by flow models [2], like detecting random noise images as OOD, what would happen when applying this method to these datasets? Besides, these OOD datasets especially random noise may not be "sharply peaked" as they are distributed widely in the input space.

[2] Input Complexity and Out-of-distribution Detection with Likelihood-based Generative Models. ICLR 2020.

**W4:** "The explanation for the OOD paradox" is a little bit weak and empirical.

**Questions:**

Here are some minor questions.

**Q1:** As claimed "While our main ideas are widely applicable to likelihood-based DGMs" in Section 4, I am wondering how this could be applicable to DGMS like diffusion models or VAEs since these two kinds of DGMs are optimizing a lower bound of the marginal data likelihood.

---

> ### Author Response · Authors · 2023-11-17
> **Response to Reviewer c1to**
>
> We thank you for your time, effort, and feedback, and we appreciate the detailed review. Here we discuss the concerns you listed. You also asked about the applicability of this method to VAEs and diffusion models; for this, please see the general rebuttal.
>
> Before addressing your concerns, we think it relevant to clarify at a high level some of the overarching themes and methodology in our work as follows.
>
> The core paradox of the study is that, while a trained DGM generates samples that look much more like its training data (e.g., CIFAR 10), it can assign higher likelihoods to data that is very clearly outside its training data distribution (eg. SVHN). Our work shares its motivation with the rich body of literature initiated by [A] and [B] which we contour in our Background and Related Work sections. Though there may exist subtle differences between generated samples and training data, the much starker dissimilitude between in-distribution/OOD pairs such as CIFAR-10/SVHN make the aforementioned paradox remarkably surprising.
>
> The notion that in-distribution points must belong to regions of higher probability mass than OOD points is broadly accepted in the literature (e.g. [A] section 2.1 and [C]) and follows naturally from first principles: if a CIFAR10 model never generates SVHN data, it must assign negligible probability to OOD regions (where OOD here is defined as SVHN). As a result, we respectfully disagree that it is a “strong assumption” without “sufficient support”.
>
> But we do not take this notion as an assumption; instead, we take it as a hypothesis (we will clarify this in the manuscript). Recall the following pair of facts outlined in our Methods section:
> - If a sample has low likelihood, it is in a region of low probability mass.
> - If a sample has high likelihood and low intrinsic dimension (i.e. the density is highly peaked), then it is also in a region of low probability mass.
>
> In our experiments (e.g. Figure 4), we see that OOD data tends to have low probability mass according to one of the above criteria, which verifies the hypothesis. We use this information to develop an OOD detection method, which as reviewer **urvk** noted, produced significant improvements over competing approaches.
>
> Having clarified our perspective, here we respond to your individual concerns:
>
>
> - (**W1**): As mentioned above, we emphasize that the stark dissimilarity between OOD and in-distribution data does make the behaviour extremely puzzling, even when models are imperfect, as evidenced by the large existing literature on the topic. We also point out that all models are imperfect (some more so than others of course), and so OOD detection with imperfect likelihood models remains a relevant problem.
>
>     We agree with you that there is “no guarantee that a deep generative model would never generate OOD samples”, and indeed since the model assigns high likelihoods to OOD data there must be some (vanishingly small) probability of sampling them. However, for all practical purposes, the well-trained models used in our examples do not generate OOD data.
>
>     Finally, we also point out that if we redefine in-distribution samples as generated samples from the (imperfect) model, OOD samples *still* achieve higher likelihoods. See for example Figure 2 (we note that this is a novel observation of ours). Because in this hypothetical situation our model is equal to $p(x_\text{IID})$, these results show that “puzzling behaviour” persists regardless of model fit.
>
> - (**W2**): Indeed several explanations have been proposed in the literature, as we summarized in the Background and Related Work sections, including [A]. In particular, the one you bring up [A] has strong assumptions, namely that the in- and out-of-distribution densities have overlapping support and that they are both Gaussian. When, as in our case, in- and out-of-distributions are semantically different, the support overlap assumption is unrealistic [C]. Our explanation and resulting method do not require either of these assumptions (note our local Taylor approximation results in a local Gaussian approximation, not a global one).

---

> ### Author Response · Authors · 2023-11-17
> **Response to Reviewr c1to (cont'd)**
>
> - (**W3**): Please see our discussion above as to why OOD samples being assigned negligible probability mass is indeed well established. We also point out that we do **not** claim OOD data is **always** assigned higher density, nor that it **always** has lower intrinsic dimension. In our work we often discus non-pathological cases like training on SVHN and evaluating on CIFAR10, where CIFAR10 data is not assigned higher likelihoods (e.g. Figure 14 for this pair). This is completely consistent with previous literature, which has found OOD data is only assigned higher likelihoods when OOD data is “simpler” than in-distribution data, which we quantify through intrinsic dimension. You also ask what would happen if we used our method in non-pathological settings. First, note that Table 1 contains several such examples, e.g. models trained on MNIST and tested on FMNIST, or trained on SVHN and tested on CIFAR10, are not subject to the standard paradox (and the intrinsic dimension of OOD data is not lower in this case), yet our method nonetheless succeeds at OOD detection. If random noise was used as OOD data instead, despite not having a low-intrinsic dimension, it would achieve a lower likelihood and still be detected by the other threshold in our dual threshold method (Algorithm 1). We will include such a result by the end of the discussion period.
> - (**W4**): We respectfully disagree with this characterization of our work. We hope that all the points we made above elucidate why, and that you will consider reevaluating our work at the end of the discussion period once we have made the promised manuscript updates. Otherwise, please let us know aspects you believe are “weak” or “empirical” and we will engage in discussion around them.
>
> [A] Nalisnick et al., “Do Deep Generative Models Know What They Don’t Know?”, ICLR 2019.
>
> [B] Choi et al., “WAIC, but Why? Generative ensembles for robust anomaly detection”, 2018.
>
> [C] Zhang et al., “Understanding Failures of Out-of-Distribution Detection with Deep Generative Models”, ICML 2021.

---

> ### Author Response · Authors · 2023-11-19
> **Experiments with Random Noise Images as OOD**
>
> We have added an extra set of experiments in response to your question in (W3) about whether our algorithm is capable of detecting random noise as OOD data because it would have high LID. In these experiments, we consider a model trained on one of the datasets in the paper and then feed it random images; each pixel intensity is chosen uniformly at random from the values 0 to 255. As you pointed out, the densities for these random images would not be sharply peaked, so they would have no low-dimensional structure and hence would have high LID.
>
> We validated our method for all datasets, and in every case the OOD datapoints are completely separated from in-distribution points by the likelihood threshold. Hence, in every case the AUC-ROC is a perfect 1.00:
>
> | In-distribution                | Out-of-distribution      | AUC - ROC |
> |------------------------|-----------------|------------------|
> | FMNIST-gen             | Random Noise    | 1.000            |
> | FMNIST                | Random Noise    | 1.000            |
> | MNIST-gen              | Random Noise    | 1.000            |
> | MNIST                 | Random Noise    | 1.000            |
> | EMNIST-gen             | Random Noise    | 1.000            |
> | EMNIST                | Random Noise    | 1.000            |
> | Omniglot-gen           | Random Noise    | 1.000            |
> | Omniglot              | Random Noise    | 1.000            |
> | CIFAR10-gen            | Random Noise    | 1.000            |
> | CIFAR10               | Random Noise    | 1.000            |
> | CelebA-gen             | Random Noise    | 1.000            |
> | CelebA                | Random Noise    | 1.000            |
> | TinyImageNet-gen       | Random Noise    | 1.000            |
> | TinyImageNet          | Random Noise    | 1.000            |
> | SVHN-gen               | Random Noise    | 1.000            |
> | SVHN                 | Random Noise    | 1.000            |
> | CIFAR100-gen           | Random Noise    | 1.000            |
> | CIFAR100              | Random Noise    | 1.000            |
>
> For a visualization of why our method is perfectly effective here, please see Figure 1 of the newly-uploaded supplementary material, which depicts estimated LID values and likelihoods for four of our models. As in the paper, each point represents a datapoint with its colour indicating whether it is in-distribution, generated from the model, or OOD. In red, we have visualized both the LID and likelihood thresholds chosen by our dual-threshold OOD detector. As illustrated, the likelihood threshold easily divides in-distribution from out-of-distribution. This is a “non-pathological” case, where the OOD datapoints lie on a flat region of the density obtaining high LID estimates while simultaneously obtaining extremely low likelihoods. The density does not peak around random noise.

---

> > ### Comment · Reviewer_c1to · 2023-11-22
> > **Thanks for your huge efforts in addressing my concerns**
> >
> > Thanks for your comprehensive response to my concerns!
> > And I appreciate the tremendous efforts in revising the paper and adding experiments to support the ideas.
> >
> > However, there are still some big concerns of mine that need further discussion.
> >
> > ----
> >
> > **For W1, sorry for my previous comment that may cause you to misunderstand my point** about "DGMs generate only in-distribution samples".
> > I do not mean that OOD samples like SVHN would be generated if you train a generative model on CIFAR-10. Actually, I agree that it is even impossible to generate data similar to SVHN though it is assigned a higher likelihood.
> > What I really care is the "samples could be very different and low-quality compared to the IID training set", an example of this can be seen in Figure 10 of your Appendix.
> >
> > Take Figure 10 (b) for example, should these generated samples be seen as "in-distribution"? If so, as you claim "in- and out-of-distributions are semantically different", what is their semantic information like category? If not, you claim "the fact that said DGMs generate only in-distribution samples" is too strong. You need to be very careful to modify this claim, otherwise, it will be abused by the following papers after acceptance.
> >
> > ---
> >
> > **Aspects I believe are “weak” or “empirical”:** I may missed something. Please allow me to check one thing, is Figure 1 (b) a real experimental result or just an illustration? If so, where are the experimental details? If not, I think that is why I think "empirical". And you may need to support your claim on such a toy example. Here is an example you may consider, see Figure 2 of this paper [1].
> > Actually, when we use a generative model to estimate the data likelihood in the high-dimension input space, where the data is actually supported on a lower dimension, the learned likelihood may be very strange to the true data likelihood, indicating that the hypothesis about "peak" maybe not the truth.
> >
> > [1] Loaiza-Ganem, Gabriel, et al. "Diagnosing and fixing manifold overfitting in deep generative models." arXiv preprint arXiv:2204.07172 (2022).
> >
> > ---
> >
> > Overall, I appreciate the authors' effort in rebuttal and I would raise my score if the above two concerns could be well addressed.

---

> > > ### Author Response · Authors · 2023-11-22
> > > **Thank you!**
> > >
> > > Thank you for engaging with us! We are glad our response has addressed most of your concerns. We address your remaining concerns below, please let us know if there is anything else we can do:
> > >
> > > 1. We agree that the phrasing “the fact that said DGMs generate only in-distribution samples” is too strong and will happily soften it. The main point we wanted to convey is that the failure of likelihood-based methods at OOD detection is highly surprising, and thus seemingly paradoxical: models are explicitly trained to assign high likelihoods to in-distribution data, yet assign higher likelihoods to (some) OOD points. This is indeed “paradoxical” in that likelihoods need to integrate to $1$, and thus one would expect that increasing likelihoods on the training distribution would decrease likelihoods on OOD points, even for an imperfect model. We have thus adjusted the wording to “the fact that said DGMs are explicitly trained to assign high likelihoods to in-distribution data without having been exposed to OOD data, thus generating samples which are much closer to the former.”
> > >
> > >     (We point out that Figure 10 (b) depicts samples from Glow, which we do not use in our experiments, but as mentioned above, we take your point that samples from an underfit model could be considered OOD in and of themselves.)
> > >
> > > 2. Figure 1 is just an illustration. We do not believe the effect illustrated in Figure 1 can necessarily be shown in a toy example though, as if the setting is too simple, the likelihoods are unlikely to exhibit pathological OOD peaks. One common observation throughout the literature is that pathologies of OOD detection are related to the relative complexity of in- and out-of-distribution data - the required complexities are hard to replicate in toy examples.
> > >
> > >     To link this to Figure 2 in [A], the authors there do show pathological behaviour of the likelihoods, but that pathological behaviour is different (although related) to the pathological behaviour resulting in OOD detection failures: in [A] the likelihood spikes to infinity on in-distribution data, and the pathology is about the rate at which the likelihood spikes to infinity within the in-distribution manifold (indeed, the toy examples in Figure 2 of [A] do not show spikes on OOD data). In contrast, the pathological behaviour driving the OOD “paradox” is that likelihoods also spike on OOD data.
> > >
> > >     We also point out that we do not see our explanation of the OOD paradox as contradictory in any way to the results in [A]. On the contrary, we believe our results are complementary: [A] shows that maximum-likelihood encourages likelihoods to diverge to infinity on the in-distribution manifold which can result in the target distribution not being properly learned. Their results do not rule out the possibility of likelihoods spiking to infinity elsewhere (since as long as they spike to infinity on the in-distribution manifold, the likelihood will be maximized). Thus, our results can be viewed as showing that empirically, these spikes do also happen elsewhere (on some OOD manifolds).
> > >
> > >     Finally, we believe that the explanation we put forth for the OOD detection paradox is not by itself “empirical”: the intuition really comes from the fact that high-dimensional densities can spike on low-dimensional manifolds while actually assigning negligible probability mass to them, as a consequence of the low-dimensionality of the manifolds (which again, is consistent with [A]). Our explanation of the OOD paradox is mathematically consistent with empirical observations, and we thus see our good empirical results at OOD detection as scientific evidence supporting a sensible hypothesis, rather than as ad-hoc or purely “empirical”.
> > >
> > > [A] Loaiza-Ganem et al., “Diagnosing and Fixing Manifold Overfitting in Deep Generative Models”, TMLR 2022

---

> > > > ### Comment · Reviewer_c1to · 2023-11-23
> > > > **Thank you for addressing my concerns!**
> > > >
> > > > **For Q1:**
> > > > >"We point out that Figure 10 (b) depicts samples from Glow, which we do not use in our experiments, but as mentioned above, we take your point that samples from an underfit model could be considered OOD in and of themselves."
> > > >
> > > > It seems that we achieved an agreement that the images in Figure 10 (b) should be considered OOD data as they have no obvious semantic information compared to the training set. with this agreement, there should be some weaknesses of this paper, firstly, you should change the overclaim "the fact" as you promised "generating samples which are much closer to the former"; secondly, the images in Figure 10 (b) seems not to lie in some OOD "peak", where your developed method may be seen as a specially designed OOD detector targeted on the "peak" OOD data, i.e., the typically used benchmark realistic image datasets.
> > > > The latter weakness exhibits a potential issue that, one can easily attack your OOD detector by an image like in Figure 10 (b).
> > > > However, as the images in Figure 10 (b) have no semantic information (at least for a human it is hard to identify any semantic information), a good OOD detector focused on semantic information should successfully detect them.
> > > >
> > > > **For Q2:**
> > > > > Figure 1 is just an illustration.
> > > >
> > > > Given Figure 1 (left) is also an illustration, I do not know why you need to put "1D" (left) plus a "2D"(right) for illustration, as they are showing the same hypotheses. Actually, Figure 1(left) could be very misleading in that one may think the OOD "peak" phenomenon actually exists in a 1-D setting. As you also claim "We do not believe the effect illustrated in Figure 1 can necessarily be shown in a toy example",
> > > > there is highly possible no such OOD peak phenomenon in a low-dimensional. So just delete Figure 1(left) may be better in the next revision.
> > > >
> > > > > if the setting is too simple, the likelihoods are unlikely to exhibit pathological OOD peaks. One common observation throughout the literature is that pathologies of OOD detection are related to the relative complexity of in- and out-of-distribution data - the required complexities are hard to replicate in toy examples.
> > > >
> > > >  This could also be added to the limitation of your method, as your method's effectiveness may be highly related to the relative complexity of in- and out-of-distribution data. As the complexity may be hard to explicitly measure though you may use some image compressor to measure it, it could be difficult to make a decision when adopting your method to achieve promising results, especially in real-world cases where we may encounter different complexity-level OOD samples.
> > > >
> > > > Overall, although there are some limitations, given the extensive analysis, the novelty of the method, and comprehensive experimental results, I believe the proposed method is a promising method to detect certain types of OOD data.
> > > >
> > > > Thus, I will hold a "borderline" attitude to this paper that I will raise my score from 3 to 5 and lower my confidence to 1.

---

> ### Author Response · Authors · 2023-11-23
> **Thank you!**
>
> We are grateful for your insightful comments and your decision to adjust your score favourably. Your feedback is invaluable, and in light of your suggestions, we will take extra care to articulate our paper more clearly, ensuring that it accurately conveys our methodology and addresses potential ambiguities.

---

### Official Review · Reviewer_urvk · 2023-10-31

**Soundness:** 3 good
**Presentation:** 4 excellent
**Contribution:** 3 good
**Rating:** 6
**Confidence:** 4

**Summary:**

This paper investigates the well known issue of probabilistic generative model assigning higher likelihood to simple OOD data. It proposes a feasible way to estimate the probability mass around a datapoint, instead of just the value of the density function. Doing so allows filtering sample with high density but low probability mass around it (i.e., sharp density function), and hence can be a reliable OOD detector. Empirical results verify the effectiveness of this method.

**Strengths:**

1. The paper tackles a well known and important issue of using DGM for OOD detection. Multiple methods has been proposed to correct the likelihood misalignment of DGM, but this paper tackles the problem in a principled way, by estimating the probability mass. Although it is intuitively clear that OOD samples with high density must have low probability mass, it is highly non-trivial to actually compute the mass. This paper shed a light on that.

2. The idea of estimating the probability mass is interesting. It smartly uses a linearization to obtained tractable Gaussian distribution, which, when doing convolution with another Gaussian distribution, results in tractable form. This is a clever design, and it's a significant technical contribution.

3. The experimental results are promising, showing significant improvements over competing methods.

**Weaknesses:**

1. The current analysis is built exclusively on normalizing flows, while it does not have discussion of other models. In particular, for models like VAEs, where the exact likelihood is not available, is it still applicable?

2. The way to determine the radius $r$ seems to be ad hoc. It depends on the dataset and the flow model as well. Is there a more principled way of choosing it? Can we just set r to be extremely small?

**Questions:**

My questions are stated above.

---

> ### Author Response · Authors · 2023-11-17
> **Response to Reviewer urvk**
>
> Thank you for your positive feedback on our ideas, methodology, and experimental results.
>
> 1. You asked whether our analysis is generalizable to models outside of normalizing flows; please see the general rebuttal for more discussion on this point.
> 2. You also asked a couple of questions about setting $r$.
>      - You asked about a more principled way of setting it; please see the general rebuttal, where we propose an improvement over our original approach.
>      - You also asked why we cannot simply set $r$ to a very small value; we agree this is a relevant question and we will further elaborate on it when updating the paper. An intuitive way to think about this is that the hyperparameter $r$ effectively chooses a scale at which we probe the density model, and that it determines how sensitive the estimate is to slight perturbations in the density. This is completely analogous to how LIDL [A] sets the various amounts of noise it uses to estimate LID, and to how commonly used estimators of global intrinsic dimension set the number of nearest neighbours as a hyperparameter [B]. Please refer to section 2.4 of [A] for a discussion on how they heuristically choose the scale variable according to how sensitive they require their estimator to be.
>
> [A] Tempczyk et al., “LIDL: Local Intrinsic Dimension Estimation Using Approximate Likelihood”, ICML 2022.
>
> [B] Levina and Bickel, “Maximum Likelihood Estimation of Intrinsic Dimension”, NeurIPS 2004.

---

> > ### Comment · Reviewer_urvk · 2023-11-23
> > **Thanks for addressing my concerns**
> >
> > My concerns on the generalization to other models as well as the way of choosing radius are addressed. However, I do think it would be better to conduct studies on at lease one more type of models (e.g., VAE), to really show the generalizability. I will maintain my score, as I think it is indeed a good contribution to OOD detection.

---

> ### Author Response · Authors · 2023-11-23
> **Thank you!**
>
> We deeply appreciate your valuable feedback. Your suggestion to illustrate the generalizability of our approach using additional model types, such as diffusions or VAEs, is well-received.

---

### Author Response · Authors · 2023-11-17
**General Response**

We thank all the reviewers for their feedback and for the time they spent reviewing our paper. We are glad that Reviewer **urvk** found our paper “clever” and that Reviewer **i5V7** enjoyed our “great visualizations” and found the paper “clearly written.” Reviewer **i5V7** also pointed to the novelty of our approach, as it is the first to leverage LID estimates for OOD detection, which Reviewer **urvk** deemed a “principled” approach. Compared to prior work, Reviewers **c1to** and **i5V7** both noted the efficiency of our method in that it requires only a single normalizing flow, which Reviewer **urvk** deemed a “significant technical contribution”. Finally, Reviewers **urvk** and **i5V7** appreciated our experimental analyses with Reviewer **urvk** noting "significant improvements" compared to the state-of-the-art.

Here we address concerns shared by at least two reviewers and reply to other questions and comments individually. We will update our manuscript towards the end of the discussion period to address all these issues as well as any other issues that may arise during the discussion.

- On our analysis focusing only on normalizing flows (**urvk** and **c1to**): As we conceded in the paper (Section 6), we agree that extending the analysis to other models will be an interesting direction for future work. We do not see our key insight, namely that intrinsic dimension provides a sensible proxy for “volume” (with “probability mass” = “volume” x “density”), as being specific to normalizing flows. Normalizing flows are simple and allow us to efficiently estimate LID and compute likelihoods, which is why we focus on them. In addition, we do not believe there is any fundamental limitation for extending our method to Gaussian VAEs, because at optimality the ELBO should be close enough to the log-likelihood (and if the lower bound nature of the ELBO remained a concern, a method like IWAE [A] could probably be leveraged) to provide a sensible approximation. We nonetheless decided not to focus on these models since better-performing VAEs tend to use discrete decoders (which do not have continuous densities) instead of Gaussian ones, and overall, we believe extending our method to work with diffusion models is likely to be the more interesting direction for future research, given their excellent empirical performance for sample generation.
- On how we set $r$ (**urvk** and **i5V7**): For our purpose, we only require a LID estimate which is good enough for OOD detection, but we do agree with our reviewers that the strategy we followed in the paper is ad-hoc and can be improved upon. To do so, we followed a new strategy leveraging LPCA [B]. LPCA is a simple LID estimator which, given an in-distribution data point, obtains its $k$-nearest neighbours, applies PCA on them, and thresholds the corresponding singular values to estimate LID. LPCA is similar to our estimator in that it uses local linearizations. In our setting, we cannot leverage LPCA to estimate LID values for OOD points, because OOD detection is unsupervised and the OOD points are not available. The only thing we assume to have access to is the training data and $p_\theta$; therefore, a very sensible and consistent method to set $r$ is to consider an $r$ where our average LID estimate aligns with LPCA’s on the training data, and then use the same $r$ to estimate LID for OOD datapoints. For expediency, we carried out this new way of setting $r$ on small subsamples of each dataset, obtaining the following ROC-AUC values:

        FMNIST (vs) MNIST: 0.9450
        MNIST (vs) FMNIST: 0.9997
        CIFAR10 (vs) SVHN: 0.9856
        SVHN (vs) CIFAR10: 0.9903

     We will include results on the full datasets by the end of the discussion period. Once again, we thank the reviewers for bringing this up, as these new results strengthen our method.


[A] Burda et al., “Importance Weighted Autoencoders”, ICLR 2016.

[B] Fukunaga and Olsen, “An Algorithm for Finding Intrinsic Dimensionality of Data”, IEEE Transactions on Computers 1971.

---

> ### Author Response · Authors · 2023-11-20
> **Principled Choice of $r$ using LPCA**
>
> As promised above, we ran experiments on every dataset pair using the new “principled” way of choosing $r$ which we described in response to the concerns of reviewers **urvk** and **i5V7**.
>
> A table containing ROC-AUCs for both Naïve Likelihoods and our method on each task is below. Each row contains two tasks  - one where generated samples from the model are considered in-distribution and one where the test data is considered in-distribution.
> The results are generally similar to those presented in the paper, with the key exceptions of (1) CIFAR-10 vs CelebA, in which our method now outperforms the best flow-based baseline, and (2) MNIST vs. Omniglot and FMNIST vs. Omniglot, in which our method achieves AUCs about 10-12% lower than our original method in the paper. We also note that in general this method tends to perform better on RGB datasets than our original method.
>
> Since setting $r$ using L-PCA performs comparably, but is done in a more principled way, we will update the manuscript to incorporate our results with this method.
>
>  | Dataset Pair:  A (and) B            | Naïve Likelihood (A-gen vs. B) | LID OOD (Ours, A-gen vs. B) | Naïve Likelihood (A vs. B)| LID OOD (Ours, A vs. B) |
> |-----------------------------|-------------------|-----------------|-------------------|-----------------|
> | EMNIST (and) MNIST          | 0.000             | 1.000           | 0.533             | 0.797           |
> | EMNIST (and) Omniglot       | 0.000             | 1.000           | 0.397             | 0.814           |
> | EMNIST (and) FMNIST         | 0.039             | 1.000           | 0.998             | 0.998           |
> | Omniglot (and) MNIST        | 0.011             | 1.000           | 1.000             | 1.000           |
> | Omniglot (and) EMNIST       | 0.000             | 1.000           | 0.983             | 0.983           |
> | Omniglot (and) FMNIST       | 0.138             | 1.000           | 1.000             | 1.000           |
> | FMNIST (and) MNIST          | 0.000             | 1.000           | 0.070             | 0.953           |
> | FMNIST (and) EMNIST         | 0.001             | 0.960           | 0.391             | 0.605           |
> | FMNIST (and) Omniglot       | 0.000             | 0.996           | 0.086             | 0.862           |
> | MNIST (and) EMNIST          | 0.000             | 1.000           | 0.985             | 0.985           |
> | MNIST (and) Omniglot        | 0.000             | 1.000           | 0.787             | 0.869           |
> | MNIST (and) FMNIST          | 0.005             | 0.999           | 1.000             | 1.000           |
> | CelebA (and) Tiny           | 0.933             | 0.965           | 0.905             | 0.936           |
> | CelebA (and) SVHN           | 0.154             | 0.930           | 0.151             | 0.949           |
> | CelebA (and) CIFAR100       | 0.946             | 0.967           | 0.919             | 0.941           |
> | CelebA (and) CIFAR10        | 0.944             | 0.965           | 0.915             | 0.935           |
> | Tiny (and) CelebA           | 0.640             | 0.646           | 0.812             | 0.812           |
> | Tiny (and) SVHN             | 0.036             | 0.951           | 0.164             | 0.907           |
> | Tiny (and) CIFAR100         | 0.686             | 0.776           | 0.796             | 0.822           |
> | Tiny (and) CIFAR10          | 0.691             | 0.767           | 0.802             | 0.825           |
> | SVHN (and) CelebA           | 0.984             | 0.984           | 0.995             | 0.995           |
> | SVHN (and) Tiny             | 0.971             | 0.971           | 0.987             | 0.987           |
> | SVHN (and) CIFAR100         | 0.970             | 0.970           | 0.987             | 0.987           |
> | SVHN (and) CIFAR10          | 0.970             | 0.970           | 0.986             | 0.986           |
> | CIFAR100 (and) CelebA       | 0.225             | 0.646           | 0.378             | 0.635           |
> | CIFAR100 (and) Tiny          | 0.392             | 0.453           | 0.485             | 0.493           |
> | CIFAR100 (and) SVHN          | 0.017             | 0.941           | 0.076             | 0.930           |
> | CIFAR100 (and) CIFAR10       | 0.402             | 0.502           | 0.492             | 0.502           |
> | CIFAR10 (and) CelebA         | 0.258             | 0.733           | 0.413             | 0.653           |
> | CIFAR10 (and) Tiny           | 0.445             | 0.555           | 0.543             | 0.548           |
> | CIFAR10 (and) SVHN           | 0.017             | 0.951           | 0.069             | 0.926           |
> | CIFAR10 (and) CIFAR100       | 0.424             | 0.610           | 0.518             | 0.561           |

---

> ### Author Response · Authors · 2023-11-23
> **Updates to the Manuscript**
>
> As promised, we have updated our manuscript, incorporating new results in response to the valuable feedback provided by the reviewers. The changes are in blue for ease of reading. We extend our sincere gratitude to all reviewers for their insightful contributions. Their input has not only enhanced the substance of our work but also improved its presentation. Please let us know if there is anything else that we can do.

---

### Meta-Review · Area_Chair_zoUL · 2023-12-07

**Metareview:**

Using likelihoods from deep generative models intuitively would seem like an ideal tool for detecting out-of-distribution data, because they are explicitly trained as a probabilistic model of the training distribution.  However, in practice the likelihood has proven to give counterintuitive results.  These models have been shown to assign high likelihoods to OOD data, but at the same time they don't generate OOD data when sampling from the model.  The authors of this paper argue that this paradox can be explained through likelihood peaks.  They argue that this can be identified using local intrinsic dimension estimation, and they develop an efficient way of doing so using normalizing flows.

The paper seems well written, well motivated and technically sound.  The reviewers found the paper quite borderline, with two 6's and 1 score of 5.  The lower score was unfortunately with the lowest possible confidence.  The reviewers found the motivation strong, seemed convinced by the method.  Their main reservations were:
* They found that the author's claim of identifying the likelihood paradox a bit too strong as it has been studied in previous literature,
* They were a bit underwhelmed that the method and experiments were limited to normalizing flows and not other generative models.

Generative models have undergone quite a resurgence recently, but unfortunately normalizing flows haven't really taken off.  It seems a shame to limit the observations of this paper to normalizing flows on small datasets like MNIST and CIFAR when there is potentially tremendous impact in dealing with OOD data in LLMs and diffusion models.  Unfortunately, the community has moved past small vision datasets, so it seems reasonable for a top-tier conference to expect experiments that are a bit more on the bleeding-edge of research.  The paper seems very close, but given the above it falls just below the bar for acceptance.  Hopefully the reviews will be helpful to make the paper stronger for a future submission.  In particular, it seems that demonstrating that the method works well on a problem of practical interest to the community would go a long way to increasing the impact of the work.

**Justification For Why Not Higher Score:**

It seems close to being there, but the experiments are too much proof-of-concept.  None of the reviewers were willing to champion the paper, and this seems reasonable when it's just hard to get excited about the experiments.

**Justification For Why Not Lower Score:**

NA

---

### Decision · Program_Chairs · 2024-01-16

Reject